# Neurovascular EGFL7 regulates adult neurogenesis in the subventricular zone and thereby affects olfactory perception

Frank Bicker[1,2], Verica Vasic[1,2], Guilherme Horta[1,2,3], Felipe Ortega[2,4,5], Hendrik Nolte[6], Atria Kavyanifar[1,2], Stefanie Keller[1,2], Nevenka Dudvarski Stankovic[1,2], Patrick N. Harter[7], Rui Benedito[8], Beat Lutz[2,4], Tobias Bäuerle[9], Jens Hartwig[10], Jan Baumgart[1,2,3], Marcus Krüger[6], Konstantin Radyushkin[1,2,3], Lavinia Alberi[11,12], Benedikt Berninger[2,4] & Mirko H.H. Schmidt[1,2]

Adult neural stem cells reside in a specialized niche in the subventricular zone (SVZ). Throughout life they give rise to adult-born neurons in the olfactory bulb (OB), thus contributing to neural plasticity and pattern discrimination. Here, we show that the neurovascular protein EGFL7 is secreted by endothelial cells and neural stem cells (NSCs) of the SVZ to shape the vascular stem-cell niche. Loss of EGFL7 causes an accumulation of activated NSCs, which display enhanced activity and re-entry into the cell cycle. EGFL7 pushes activated NSCs towards quiescence and neuronal progeny towards differentiation. This is achieved by promoting Dll4-induced Notch signalling at the blood vessel-stem cell interface. Fewer inhibitory neurons form in the OB of *EGFL7*-knockout mice, which increases the absolute signal conducted from the mitral cell layer of the OB but decreases neuronal network synchronicity. Consequently, *EGFL7*-knockout mice display severe physiological defects in olfactory behaviour and perception.

[1] Molecular Signal Transduction Laboratories, Institute for Microscopic Anatomy and Neurobiology, Johannes Gutenberg University, School of Medicine, Langenbeckstraße 1, 55131 Mainz, Germany. [2] Focus Program Translational Neuroscience (FTN), Rhine Main Neuroscience Network (rmn²), Johannes Gutenberg University, School of Medicine, Langenbeckstraße 1, 55131 Mainz, Germany. [3] Translational Animal Research Center (TARC), Johannes Gutenberg University, School of Medicine, Hanns-Dieter-Hüsch-Weg 19, 55128 Mainz, Germany. [4] Institute of Physiological Chemistry, Johannes Gutenberg University, School of Medicine, Duesbergweg 6 & Hanns-Dieter-Hüsch-Weg 19, 55128 Mainz, Germany. [5] Biochemistry and Molecular Biology Department IV, Faculty of Veterinary Medicine, Complutense University, Av Puerta de Hierro s/n, 28040 Madrid, Spain. [6] Institute for Genetics and Cologne Excellence Cluster on Cellular Stress Responses in Aging-Associated Diseases (CECAD) and Center for Molecular Medicine (CMMC), University of Cologne, Joseph-Stelzmann-Straße 26, 50931 Cologne, Germany. [7] Edinger-Institute (Neurological Institute), Goethe-University, School of Medicine, Heinrich-Hoffmann-Str. 7, 60528 Frankfurt am Main, Germany. [8] Molecular Genetics of Angiogenesis Laboratory, Centro Nacional de Investigaciones Cardiovasculares, Melchor Fernández Almagro 3, 28029 Madrid, Spain. [9] Preclinical Imaging Platform Erlangen, Institute of Radiology, University Medical Center Erlangen, Palmsanlage 5, 91054 Erlangen, Germany. [10] Institute for Molecular Biology, Johannes Gutenberg University, Ackermannweg 4, 55128 Mainz, Germany. [11] Swiss Integrative Center for Human Health SA, Passage du Cardinal 13B, CH-1700 Fribourg, Switzerland. [12] Unit of Pathology, Department of Medicine, University of Fribourg, Rte Albert Gockel 1, CH-1700 Fribourg, Switzerland. Correspondence and requests for materials should be addressed to M.H.H.S. (email: mirko.schmidt@unimedizin-mainz.de).

Neural stem cells (NSCs) reside in a specialized micro-environment that maintains them as undifferentiated quiescent cells (qNSCs) but allows them to re-enter the cell cycle upon stimulation and become activated (aNSCs) to sustain life-long regeneration of the neuronal population[1–3]. The subventricular zone (SVZ) is one of two germinal niches of the adult mammalian brain where NSCs reside and give rise to new neurons throughout life. This process of neurogenesis is initiated from type B qNSCs that, upon activation, turn into aNSCs and give rise to type C transit-amplifying progenitors (TAPs). TAPs become type A neuroblasts (NBs) and migrate along the rostral migratory stream (RMS) to the olfactory bulb (OB) where they eventually differentiate into inhibitory interneurons[4]. In the mouse, function and structure of the OB heavily rely on this constant influx of adult-born neurons, which form various types of interneurons[5] and contribute to neural plasticity of olfactory information processing and pattern discrimination[6]. Adult NSCs display a characteristic morphology that allows them to bridge the three layers of the SVZ[7]. NSCs come into contact with the cerebrospinal fluid by an apical process, with their progeny by their intermediate segment and associate with blood vessels by a basal endfoot[8].

The molecular cues governing NSC maintenance, neurogenesis and the role of blood vessels in this context have not been fully explored but it has been demonstrated that NSCs exist in a specialized vascular niche, where the regulation of NSCs in SVZ relies on the local vasculature[9,10] and the cerebrospinal fluid[11]. However, only a few blood vessel-derived factors regulating NSCs have been identified, such as VEGF[12,13], NT-3 (ref. 14), betacellulin[15], PEDF[16], PlGF-2 (ref. 17) or the Notch ligand Jagged1 (ref. 18). Notch pathway components are broadly expressed throughout the brain and are particularly active in SVZ NSCs[4,19,20]. Notch receptors share a common architecture and are activated by canonical Delta- or Jagged-type ligands by specific EGF repeats by a series of cleavage events, which eventually release the Notch intracellular domain and activates the transcription factor RBPJ[21]. A putative blood vessel-derived Notch modulator acting in this context is the epidermal growth factor-like protein 7 (EGFL7), which has been identified as a non-canonical Notch ligand and inhibitor of Jagged-induced Notch signalling in NSCs in vitro[22]. Further, EGFL7 has been shown to be secreted by neurons and blood vessels in the adult brain[23,24].

In this work the potential of EGFL7 to act as an angioneurin governing NSCs in vivo is elicited. We show where the protein is localized in the SVZ, how it governs NSCs in vivo and unravel its function for olfactory perception in the OB

## Results

### Localization of EGFL7 expression in the mouse brain and SVZ.
To define the temporal and spatial expression of EGFL7 in embryonic development, RNA was isolated from murine embryos at various stages ranging from embryonic day 9 (E9) to E18 as the latest stage. The expression levels of EGFL7 were compared by quantitative reverse transcriptase-polymerase chain reaction (qRT-PCR). EGFL7 expression peaked at E9 (Fig. 1a), when embryonic blood vessel formation reaches a maximum and was restricted to vascular structures (Fig. 1b–d and Supplementary Fig. 1a–i). Another wave of EGFL7 expression was observed at E18 (Fig. 1a) when the first non-vascular cells were spotted in the marginal zone of the developing brain (Supplementary Fig. 1j–l). Morphology, shape and local distribution identified these cells as Cajal–Retzius neurons (Supplementary Fig. 1k). In late prenatal and various postnatal stages a continuously rising level of EGFL7 was observed, lasting until adulthood at postnatal stage P40 (Fig. 1e). Subsequently, the expression levels declined but

remained stable at significant levels until high age. In situ and fluorescence in situ hybridization (FISH) revealed the overall high expression of EGFL7 in neurons throughout the brain, for example, in the cortex (Fig. 1f–h) but lower EGFL7 levels in the SVZ (Supplementary Fig. 1m–q). Nevertheless, EGFL7 was detected in vascular (Supplementary Fig. 1n–p, arrowheads) and non-vascular (Supplementary Fig. 1n–p, asterisks) structures. Interestingly, the RMS (Supplementary Fig. 1r, region inside the dotted line), which NBs utilize to exit the SVZ and to migrate towards differentiation into the OB, was fully devoid of an EGFL7 signal, while the striatum surrounding the RMS (Supplementary Fig. 1r, region surrounding the dotted line) and the OB (Supplementary Fig. 1s,t) displayed distinct EGFL7 signals.

EGFL7-expressing cells in the SVZ were visualized by a combination of FISH, using EGFL7-specific probes, and immunofluorescence (IF) to identify specific cell types. Strong EGFL7 expression was detected in ECs (Fig. 1i,j, arrowheads) and aNSCs/NPCs (Fig. 1k,l, arrowheads). In addition, EGFL7 expression was sporadically detected in TAPs (Supplementary Fig. 1u) and in neuronal projections (Supplementary Fig. 1v). NBs (Supplementary Fig. 1w), Eps (Supplementary Fig. 1x) and astrocytes (Fig. 1l, arrows; Supplementary Fig. 1y) stained negative for EGFL7.

To quantitatively compare EGFL7 levels in different SVZ cell types, these were isolated by fluorescence-activated cell sorting (FACS). qRT-PCR revealed that ECs and qNSCs synthesized particularly high levels of EGFL7 (Fig. 1m). Smaller amounts were traced in aNSCs and TAPs, but no expression was detected in NBs, Eps or astrocytes. Considering the excess of ECs over NSCs (>10:1 as determined by FACS sorting), blood vessels seem to be the major source of EGFL7 in the SVZ.

### EGFL7 affected SVZ-derived neurospheres.
The functional impact of EGFL7 on NSCs/NPCs in vitro was analysed in SVZ-derived neurospheres by assessment of the amount of secondary formed spheres (self-renewal) and sphere size. An adenoviral approach was applied to ectopically express EGFL7 in infected spheres at an early phase of sphere formation (Fig. 2a). Comparable viral infection rates were monitored by enhanced green fluorescent protein (EGFP) expression and FACS (Supplementary Fig. 2a,b). Furthermore, it has been verified that neurosphere assays performed at clonal density or under clonal conditions upon FACS sort yielded comparable results (Supplementary Fig. 2c). The self-renewal potential of neuro-spheres was significantly reduced upon the ectopic expression of EGFL7 using AdEGFL7 (Supplementary Fig. 2d). Further, EGFL7-infected spheres were smaller (Fig. 2b; <20 μm) and proliferated less as quantified by qRT-PCR of the proliferation markers Ki67 and MCM2 (Supplementary Fig. 2e) as previously described[22]. Data indicate that EGFL7-infected spheres failed to overcome the single/few cell state. Conversely, $EGFL7^{-/-}$ spheres displayed an increased self-renewal potential (Supplementary Fig. 2f), confirming previous knockdown approaches[22]. Furthermore, $EGFL7^{-/-}$ spheres grew larger (Fig. 2c; >75 μm) and displayed enhanced proliferation (Supplementary Fig. 2g).

Proteomics analyses of $EGFL7^{-/-}$ and wild-type (WT) neurospheres by SILAC-based mass spectrometry[25] at day (d) 2 and d5 post seeding, using Lys(6)-labelled mouse lysates as an internal spike-in protein standard (Supplementary Fig. 2h), allowed for the relative protein quantification of more than 4,000 proteins (Supplementary Data 1: XPro). Neurosphere protein profiles at d2 (WT, $EGFL7^{-/-}$) and d5 ($EGFL7^{-/-}$) were comparable (Supplementary Fig. 2i) and indicative of enhanced metabolism and proliferation. Comparison of log2 ratio distributions of $EGFL7^{-/-}$ and WT at d2 and d5 by a Violin

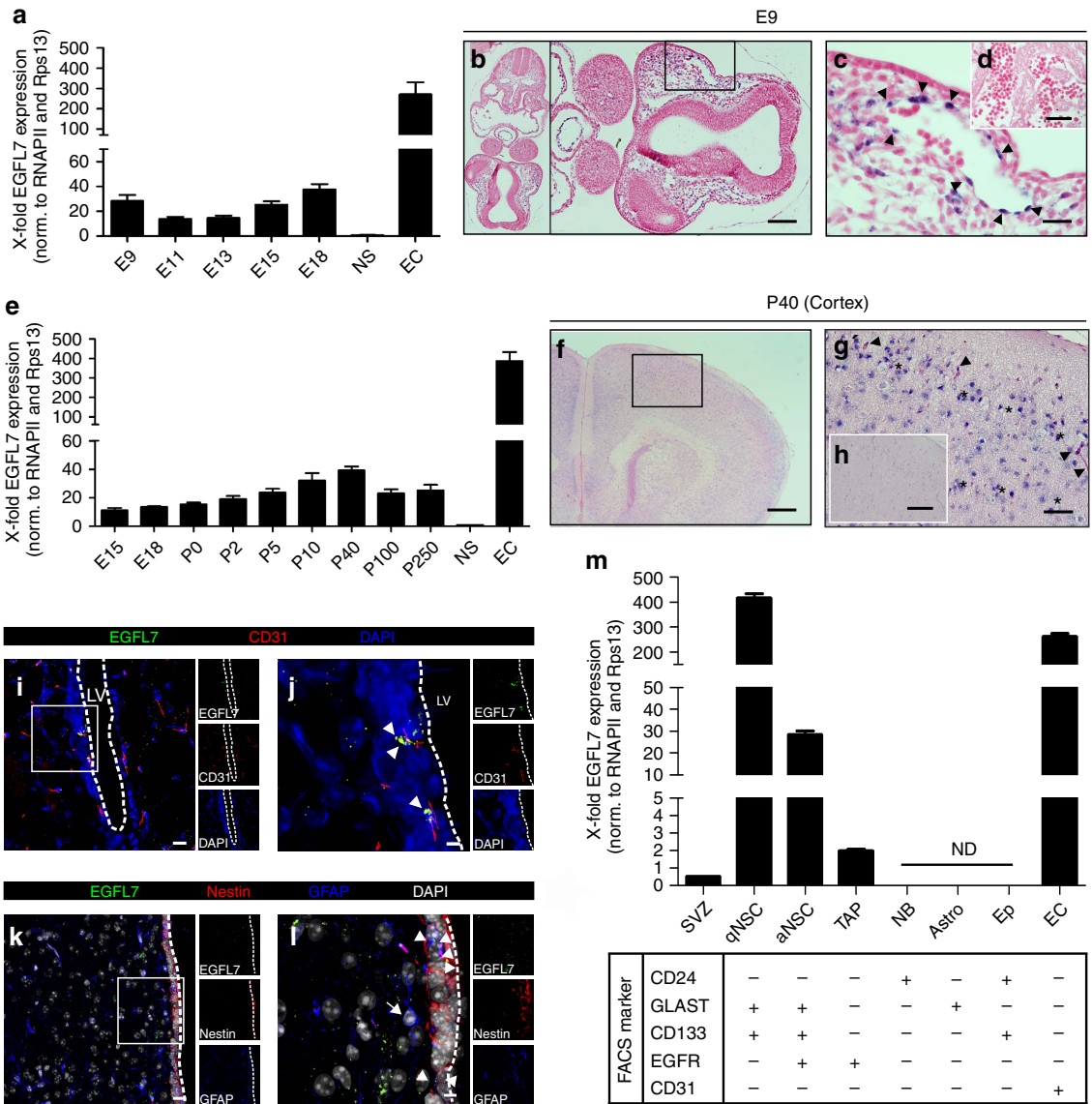

**Figure 1 | Localization of EGFL7 expression in the mouse brain and SVZ.** (**a**) Two expression maxima of EGFL7 were detected at E9 and E15/E18 by qRT-PCR. EC, endothelial cells, positive control; NS, neurospheres, low level reference. (**b**) *In situ* hybridization of EGFL7, E9 mouse embryo. (**c**) Higher magnification of the inset in (**b**). EGFL7 was expressed in ECs (arrowheads). (**d**) Negative control of (**b**). (**e**) Continuous EGFL7 expression measured in the mouse brain by qRT-PCR until high age with a maximum at P40. (**f**) *In situ* hybridization of EGFL7, P40 cortex. (**g**) Higher magnification of the inset in (**f**). EGFL7 was expressed in ECs (arrowheads) and non-ECs (asterisks). (**h**) Negative control of (**f**). (**i–l**) The following markers were applied to discriminate among different cell types by FISH and IF analysis: CD31 for endothelial cells (ECs), Nestin and GFAP for aNSCs or GFAP for astrocytes. (**j,l**) Higher magnification of the inset in (**i,k**) EGFL7 expression was majorly detected in ECs (**j**, arrowheads) and aNSCs/NPCs (**l**, arrowheads). Astrocytes stained negative for EGFL7 (**l**, arrow). (**m**) SVZ cell populations were FACS-purified using the following combinations of markers for discrimination: CD31$^+$ for ECs, GLAST$^+$/CD133$^+$/EGFR$^-$ for qNSCs, GLAST$^+$/CD133$^+$/EGFR$^+$ for aNSCs, CD133$^-$/EGFR$^+$ for TAPs, CD24$^+$ for NBs, CD24$^+$/CD133$^+$ for Eps and GLAST$^+$/CD133$^-$ for astrocytes. qRT-PCR revealed high EGFL7 expression in qNSCs and ECs, moderate expression in aNSCs and TAPs but no expression in NBs, Eps and astrocytes (ND). LV, lateral ventricle. Scale bar, 500 μm (**f**), 200 μm (**h**), 100 μm (**b,g**), 50 μm (**d**), 25 μm (**c,i,k**) and 5 μm (**j,l**).

plot (Supplementary Fig. 2j) revealed a wider distribution of proteins at d5, indicating distinct molecular signatures of $EGFL7^{-/-}$ and WT spheres. Differential protein expression using gene ontology (GO) annotations and one-dimensional enrichment displayed 317 distinct GO categories at a false discovery rate (FDR) below 0.01 (Supplementary Data 2: XGO) and revealed a lack of synaptic proteins in $EGFL7^{-/-}$ spheres at d5 (Fig. 2d, Supplementary Fig. 2k). Moreover, proteins affiliated with glutamate receptors, potassium channels, synaptic vesicles and juxtaparanode regions of axons were two- to five-fold downregulated, suggesting $EGFL7^{-/-}$ retained maturation of neurosphere cells and kept aNSCs in a proliferative mode.

Limited differentiation paralleled by retained proliferation of $EGFL7^{-/-}$ spheres was further analysed by a cell cycle analysis, which revealed an increased amount of cells in $G_2/M$ phase at the expense of cells in $G_0/G_1$ phase (Fig. 2e). An increased amount of neurospheres was recovered from the aNSC and TAP populations upon FACS-purification of SVZ-cells (Fig. 2f indicating that in particular the sphere formation capacity of aNSCs was increased in $EGFL7^{-/-}$ mice.

Continuous live imaging of primary $EGFL7^{-/-}$ and WT cultures was performed to track proliferation and lineage progression of NSCs/NPCs *in vitro* (Fig. 2g–r) in the absence of growth factors. Time-lapse video microscopy of primary

NSC/NPC cultures was performed by a cell observer for 5–7 d. Under these culture conditions, aNSCs exhibited a prototypical lineage tree (Supplementary Movie 1) and recapitulated NSC differentiation *in vitro*[26]. Subsequent to acquiring the characteristic morphology of quiescent astroglial cells, $EGFL7^{-/-}$

NSCs re-entered the cell cycle (Fig. 2k–m and Supplementary Movie 2) and re-started differentiation, an effect never observed before with WT cells. Data suggest loss of EGFL7 increased the amount of proliferating aNSCs/NPCs by the reactivation of quiescent cells.

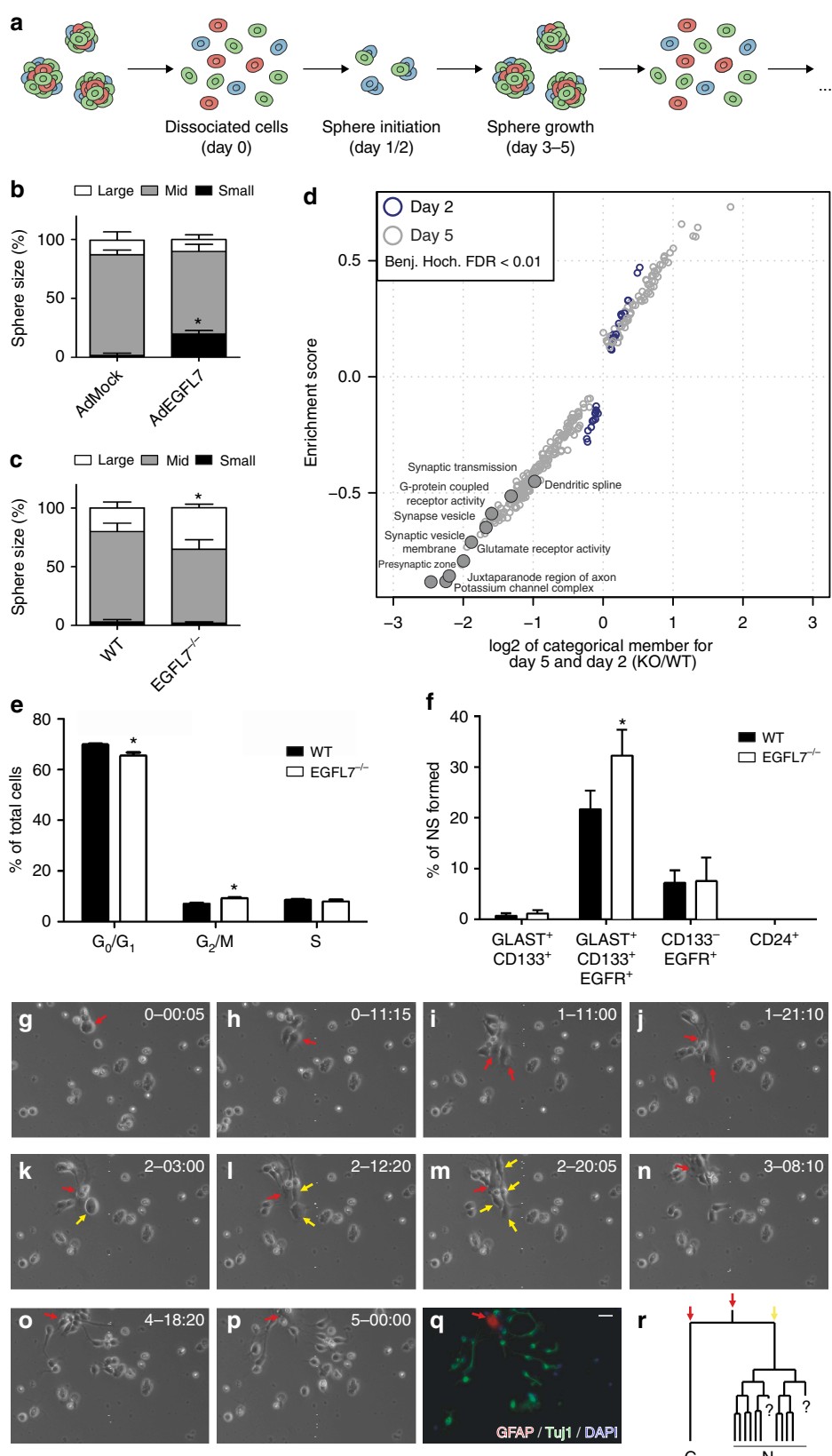

**EGFL7 modified Notch signalling in the SVZ.** Previously, EGFL7 has been identified as a non-canonical Notch ligand that affects Notch signalling *in vitro*. This finding was verified by the infection of neurospheres with AdEGFL7 (an adenovirus encoding for EGFL7), which caused a reduction in the relative amount of the Notch reporter gene Hes5 (Fig. 3a) as previously presented[22]. However, cerebroventricular injection (CVI) of AdEGFL7 to induce expression of EGFL7 in the SVZ, followed by microdissection of the SVZ, RNA isolation and qRT-PCR, revealed elevated levels of Hes5 upon ectopic EGFL7 expression (Fig. 3b). Conversely, Hes5 levels were elevated in $EGFL7^{-/-}$ neurospheres (Fig. 3c) and total Hes5 levels were decreased in the SVZ of $EGFL7^{-/-}$ mice (Fig. 3d). Two-coloured multiplex FISH in WT SVZ tissue, combining EGFL7 and Hes5 *in situ* probes, revealed that Hes5 levels close to cells expressing EGFL7 were higher as compared to regions without EGFL7 in the SVZ (Fig. 3e,f). Micrographs were subjected to Imaris-based quantification and cross-correlation analysis. Regions staining positive for EGFL7 yielded stronger signals for Hes5 as compared to regions with low EGFL7 levels (Fig. 3f,g).

To define potential candidates responsible for the differential effects of EGFL7 *in vitro* and *in vivo*, Notch signalling components in SVZ and neurospheres were quantified by qRT-PCR. Prominent levels of the Notch receptors 1–3, the canonical ligands Jagged1 and Dll1 as well as the Notch reporter genes Hes1 and Hes5 were spotted in both tissues (Supplementary Fig. 3a,b), but Notch4, Jagged2 and Dll3 were mostly absent. Notch3, Jagged1, Dll1 and Hes1 displayed markedly increased expression levels in neurospheres as compared to the SVZ, while Notch2 was slightly reduced. Interestingly, the Notch ligand Dll4 displayed a differential expression pattern and was readily detectable in SVZ tissue but not in neurospheres. *In vivo* localization of these Notch signalling components was performed by two-coloured multiplex FISH, combining an EGFL7 *in situ* probe with probes for Notch1–3, Jagged1, Dll1 or Dll4. Data revealed an ubiquitous expression of Notch1–3 in the SVZ (Fig. 3h,i and Supplementary Fig. 3c,d) as previously described[27]. Co-localization with EGFL7 occurred in individual hotspots and often within the same cell. Expression of the Notch ligands Jagged1, Dll1 and Dll4 was not as ubiquitous compared to the Notch receptors but displayed a more punctuated pattern with individual hotspots colocalizing with EGFL7, usually in neighbouring cells (Fig. 3j,k and Supplementary Fig. 3e,f). Interestingly, IF staining of Dll4 revealed its prominent expression in blood vessels (Supplementary Fig. 3g), where it readily colocalized with EGFL7 (Supplementary Fig. 3h).

To investigate the influence of ectopically expressed Dll4 *in vivo*, CVI of adenovirus encoding for Dll4 (AdDll4) into the ventricle of WT mice was performed. IF and quantification of aNSCs and their progeny revealed a decreased amount of proliferating aNSCs/NPCs (Fig. 3l) and TAPs (Fig. 3m) in the SVZ of AdDll4-infected animals. Conversely, the induced blood vessel-specific knockout of Dll4 in $Dll4^{i\Delta EC}$ mice, created by treatment of $Dll4^{fl/fl};Cdh5(PAC)$-CreERT2 animals with tamoxifen, increased the amount of proliferating aNSCs/NPCs (Fig. 3n) and TAPs (Fig. 3o). The amount of primary neurospheres, forming from microdissected SVZ tissue of these animals *ex vivo*, was increased upon Dll4 knockout in blood vessels (Fig. 3p). Remarkably, the stem cell-specific knockout of Dll4 using a $Dll4^{fl/fl};Nestin$-CreERT2 model ($Dll4^{i\Delta NSC}$) did not affect primary sphere formation (Supplementary Fig. 3i), suggesting negligible autocrine Dll4-mediated Notch signalling within the NSC population. The amount of these neurosphere initiating cell-derived spheres (NICs) *ex vivo* served as a measure for the amount of aNSCs *in vivo*. Injection of AdEGFL7 into the ventricle of $Dll4^{fl/fl};Cdh5(PAC)$-CreERT2 animals reduced the amount of NICs by about 40% (Fig. 3p) but this phenotype was reduced upon Dll4 knockout in blood vessels (ca. − 25%). Conversely, injection of AdDll4 into the ventricle of $EGFL7^{-/-}$ mice (Fig. 3q) reduced the amount of NICs (ca. − 25%), however, to a lesser extent as compared to WT controls (ca. − 35%). Data suggest endothelial Dll4 and EGFL7 act as cooperative regulators of NSC activity.

Indeed, Dll4 and EGFL7 induced Notch signalling in a cooperative manner as measured by qRT-PCR of the Notch target gene Hes5 in neurospheres seeded on plastic dishes coated with recombinant purified proteins (Supplementary Fig. 3j). Furthermore, knockdown of Dll4 or EGFL7 in human umbilical vein endothelial cells (HUVECs) using specific siRNAs reduced the amount of Notch signalling in the co-cultured murine NSCs in an additive manner as measured by qRT-PCR of murine Hes5 (Supplementary Fig. 3k). To determine the molecular mechanism behind these observations, the interactions of Notch1 with its ligands EGFL7, Dll4 and Jagged1 were studied. Flag-tagged EGFL7 and the V5-tagged extracellular domains of all four Notch receptors were co-expressed in HEK293 cells. Subsequent co-immunoprecipitation (co-IP) analyses followed by quantitative western blot using an Odyssey system revealed that EGFL7 bound strongest to the ECD of Notch1 but with declining strength to Notch2 (ca. 76%), Notch3 (ca. 41%) and Notch4 (ca. 35%) (Supplementary Fig. 3l; $n = 3$; $P < 0.05$). Subsequently, the EGFL7 binding site within Notch1 was mapped using Flag-tagged EGFL7 along with V5-tagged Notch1-ECD deletion mutants. These constructs lacked different EGF-like repeats that are relevant for Notch signalling. Co-IPs revealed reduced binding of EGFL7 to Notch1 mutants lacking EGF-like repeats relevant for Notch1 activation *in trans* (paracrine) (Supplementary Fig. 3m; Notch1-Δ5–10 (ca. − 18%), Notch1-

**Figure 2 | EGFL7 affected SVZ-derived neurospheres.** (**a**) Scheme of neurosphere assay. (**b**) Ectopic EGFL7 expression increased the amount of small neurospheres (19.7 ± 3.31 versus 1.54 ± 1.34% in AdMock; $n = 100$; $P < 0.01$), while (**c**) $EGFL7^{-/-}$ spheres displayed an increased size (35.3 ± 2.98 versus 20.15 ± 5.23% in WT; $n = 100$; $P < 0.01$). (**d**) SILAC-based mass spectrometric proteome analyses of $EGFL7^{-/-}$ and WT neurospheres was performed after 2 and 5 d in culture. WT spheres upregulated proteins involved in neuronal signalling, for example, glutamate receptors or synaptic proteins at d5, suggesting differentiation *in vitro*. However, $EGFL7^{-/-}$ spheres displayed limited upregulation of these proteins but instead, retained an expression profile at d5 that was comparable to WT and knockout (KO) spheres at d2 in culture with high levels of riboproteins and metabolic enzymes. This indicates that $EGFL7^{-/-}$ spheres displayed delayed differentiation but sustained proliferation *in vitro* resulting in larger spheres. (**e**) Cell cycle analysis revealed an increased amount of cells in the $G_2/M$ phase (9.08 ± 0.62 versus 6.91 ± 0.61% in WT; $n = 3$; $P < 0.05$) at the expense of cells in $G_0/G_1$ phase (65.36 ± 2.40 versus 69.76 ± 0.80% in WT; $n = 3$; $P < 0.05$). (**f**) FACS sorting of qNSCs, aNSCs, TAPs and NBs from the SVZ of $EGFL7^{-/-}$ and WT mice was performed using the following markers: qNSCs (GLAST$^+$/CD133$^+$/EGFR$^-$), aNSCs (GLAST$^+$/CD133$^+$/EGFR$^+$), TAPs (CD133$^-$/EGFR$^+$) and NBs (CD24$^+$). Subsequently, an *in vitro* neurosphere formation assay yielded an increased amount of spheres derived from the aNSC population (32.3 ± 5.1 versus 21.7 ± 3.7% in WT; $n = 3$; $P < 0.05$). (**g–r**) Time-lapse video microscopy analysis of primary NSC/NPC cultures. (**g–p**) Individual time points of the experiment. (**q**) Post hoc cell identification by IF. Scale bar, 50 μm. (**r**) A typical differentiation tree of $EGFL7^{-/-}$ aNSCs displays initial cell divisions (red arrows) and differentiation but subsequently, re-entering of cell cycle (yellow arrows) and further differentiation, which is not observed in WT. Statistical analysis was performed by Student's *t*-test/Mann–Whitney *U*-test. For all experiments data, error bars indicate s.d.

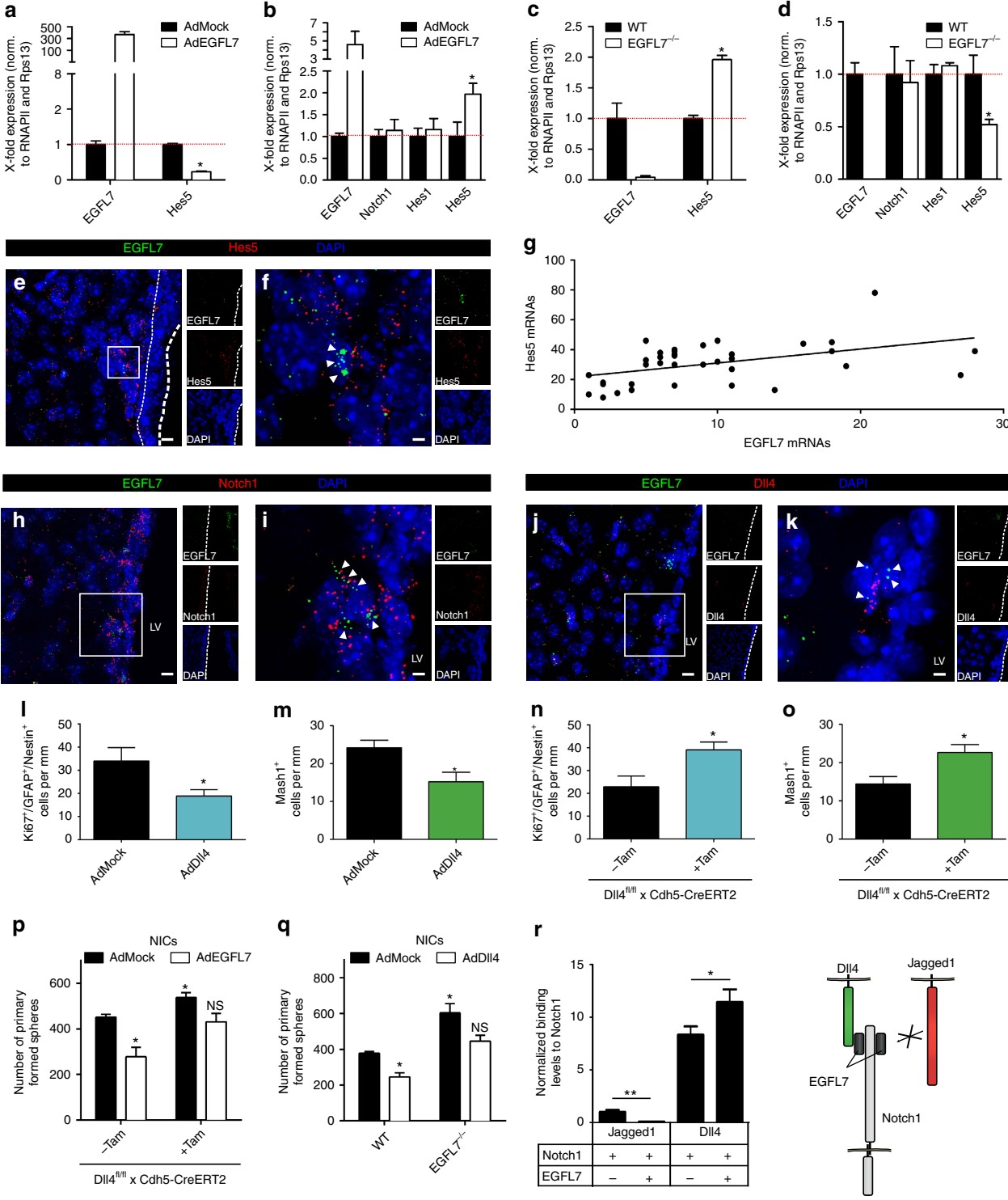

**Figure 3 | EGFL7 modified Notch signalling in the SVZ. (a)** Ectopic EGFL7 reduced Hes5 in neurospheres *in vitro*[22] (0.23 ± 0.01 versus 1 ± 0.02-fold in AdMock; *n* = 3; *P* < 0.001) **(b)** but increased it in SVZ tissue *in vivo* (1.97 ± 0.25 versus 1 ± 0.33-fold in AdMock; *n* = 3; *P* < 0.05). **(c)** Conversely, Hes5 was upregulated in *EGFL7*[−/−] spheres (1.96 ± 0.04 versus 1 ± 0.03-fold in WT; *n* = 3; *P* < 0.01) but **(d)** downregulated in the SVZ of *EGFL7*[−/−] mice (0.52 ± 0.05 versus 1 ± 0.18-fold in WT; *n* = 3; *P* < 0.005). **(e,f)** FISH revealed co-localization of Hes5 and EGFL7 mRNAs in the SVZ (arrowheads), **(g)** quantified by cross-correlation analysis (*r* = 0.46; *P* < 0.005). **(h–k)** EGFL7 and Notch1 transcripts were spotted in close vicinity or within the same cell **(h,i,** arrowheads) while EGFL7 and Dll4 were found exclusively in neighbouring cells **(j,k,** arrowheads). **(f,i,k)** Higher magnifications. LV, lateral ventricle. Scale bar, 25 μm **(e,h,j)** or 5 μm **(f,i,k)**. **(l)** Ectopic Dll4 *in vivo* reduced aNSCs (17.79 ± 1.08 versus 35.29 ± 6.99 cells per mm in AdMock; *n* = 3; *P* < 0.05) and **(m)** TAPs (15.22 ± 2.52 versus 24.18 ± 2.01 cells per mm in AdMock; *n* = 3; *P* < 0.05). Conversely, Dll4-deficient blood vessels increased **(n)** aNSCs (45.33 ± 4.12 versus 22.95 ± 4.66 cells per mm in WT; *n* = 3; *P* < 0.05) and **(o)** TAPs (39.11 ± 3.41 versus 28.80 ± 3.99 cells per mm in WT; *n* = 3; *P* < 0.05) in the SVZ. *Dll4*[iΔEC] mice **(p)** gave rise to more primary neurospheres *ex vivo* (NICs; 503.67 ± 21.55 versus 452.5X ± 7.78 spheres in WT; *n* = 3; *P* < 0.05), an effect partially reversible by AdEGFL7. **(q)** *EGFL7*[−/−] yielded more NICs *ex vivo* but AdDll4 could not fully rescue this phenotype. **(r)** Biochemical analyses revealed competition between EGFL7 and Jagged1 but cooperation with Dll4 in terms of Notch1 binding (*n* = 3; *P* < 0.05). Statistics performed using Student's *t*-test/Mann–Whitney *U*-test. Error bars indicate s.d.

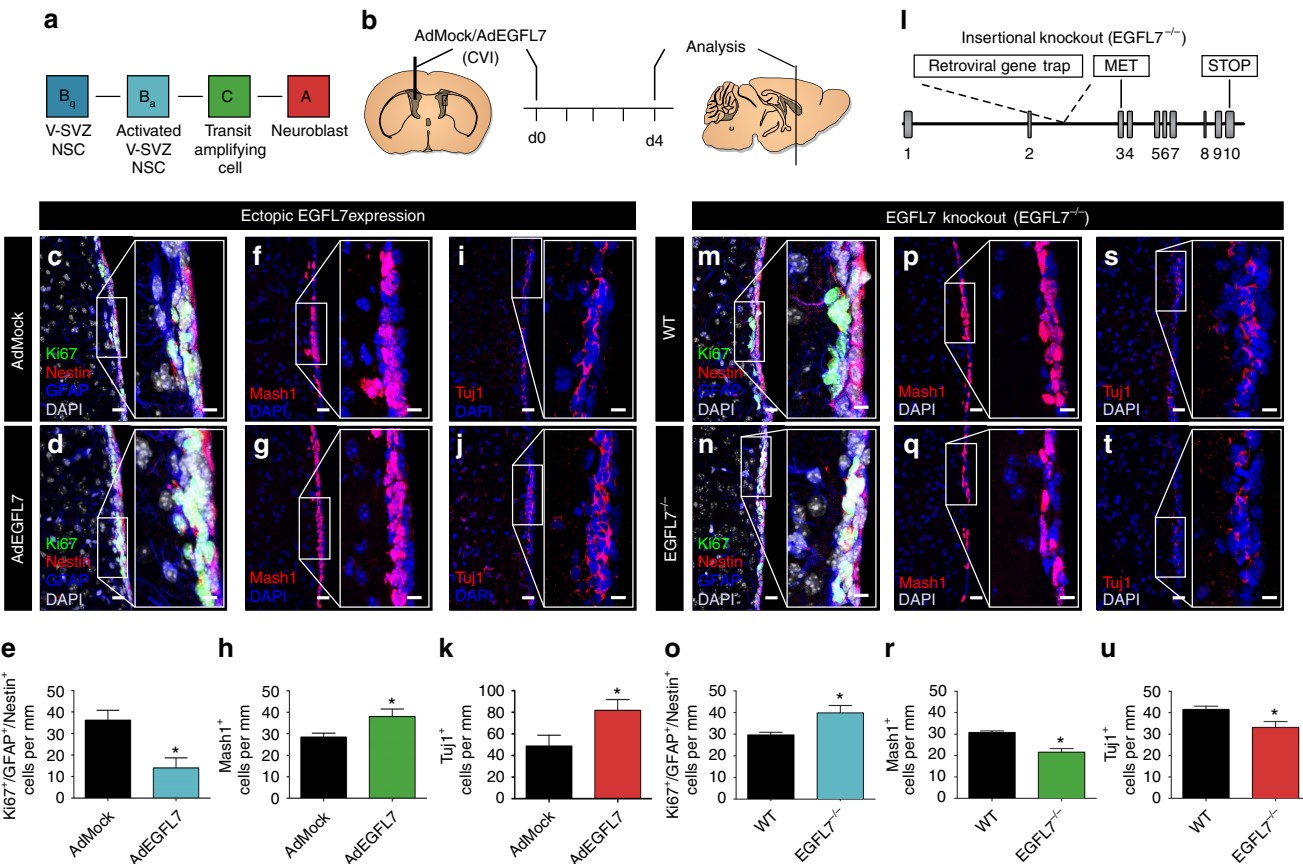

**Figure 4 | EGFL7 modulated the cytoarchitecture of the SVZ.** (**a**) Prototypical model of NSC differentiation in the SVZ. (**b**) Schematic representation of the experimental set-up for CVI. Quantitative IF analyses of the SVZ cytoarchitecture upon (**b–k**) ectopic EGFL7 expression or (**l–u**) $EGFL7^{-/-}$ using markers for (**c,d,m,n**) aNSCs/NPCs, (**f,g,p,q**) TAPs and (**i,j,s,t**) NBs revealed (**e**) a significant decrease in aNSCs/NPCs ($13.97 \pm 4.73$ versus $36.13 \pm 4.67$ cells per mm in AdMock; $n=6$; $P<0.05$), but (**h**) an increase in TAPs ($38.07 \pm 3.46$ versus $28.47 \pm 1.83$ cells per mm in AdMock; $n=6$; $P<0.05$) and (**k**) NBs ($81.78 \pm 10.11$ versus $48.76 \pm 10.14$ cells per mm in AdMock; $n=6$; $P<0.05$) upon ectopic expression of EGFL7. $EGFL7^{-/-}$ caused an (**o**) increase in aNSCs ($34.03 \pm 0.04$ versus $31.32 \pm 1.15$ cells per mm in WT; $n=6$; $P<0.05$) but (**r**) a decrease in TAPs ($21.55 \pm 1.66$ versus $30.75 \pm 0.76$ cells per mm in WT; $n=6$; $P<0.005$) and (**u**) NBs ($33.1 \pm 2.78$ versus $41.5 \pm 1.52$ cells per mm in WT; $n=6$; $P<0.05$). Data suggest EGFL7 pushed aNSCs towards differentiation and $EGFL7^{-/-}$ caused an accumulation of aNSCs. Statistical analysis was performed by Student's $t$-test/Mann–Whitney $U$-test. For all experiments data, error bars indicate s.d. Scale bar, 25 µm (**c–t**) and 10 µm (magnifications).

$\Delta11+12$ (ca. $-26\%$) and Notch1-$\Delta8$ (ca. $-28\%$); $n=3$; $P<0.05$). Remarkably, Notch1-$\Delta24$–$29$, lacking a region responsible for *in cis* (autocrine) inactivation of Notch1, was not significantly impaired in EGFL7 binding. This indicates that EGFL7 may interfere with Notch activation *in trans* without affecting inactivation *in cis*. Furthermore, the consequences of EGFL7 expression for the binding of Notch1 to its ligands Jagged1 and Dll4 were studied. V5-tagged Notch1-ECD and Flag-tagged ligands were co-expressed in HEK293 in the absence or presence of EGFL7. Co-IPs revealed that EGFL7 reduced Notch1 binding to Jagged1 (ca. $-92\%$) but increased binding to Dll4 (ca. $+37\%$) (Fig. 3r, Supplementary Fig. 3n). Thereby EGFL7 promoted Notch1 activation induced by Dll4 at the expense of Jagged1.

**EGFL7 modulated the cytoarchitecture of the SVZ.** To quantify the impact of EGFL7 on adult neurogenesis in the SVZ *in vivo*, aNSCs and their progeny were quantified by IF (Fig. 4a). Mice underwent CVI of AdEGFL7 to ectopically express EGFL7 in the SVZ, AdMock served as a negative control (Fig. 4b). Effective infection *in vivo* was verified by the detection of virus-encoded EGFP and anti-EGFL7 IF staining (Supplementary Fig. 4a–e). AdEGFL7 did neither affect the total number of aNSCs/NPCs

(Supplementary Fig. 4f) nor apoptosis as detected by TUNEL staining (Supplementary Fig. 4g–i). IF of the proliferation marker Ki67 revealed a decreased amount of proliferating aNSCs/NPCs *in vivo* on CVI of AdEGFL7 (Fig. 4c–e). Further, AdEGFL7 caused an increase in the amount of TAPs (Fig. 4f–h) and NBs (Fig. 4i–k). IF analysis of $EGFL7^{-/-}$ and WT tissue (Fig. 4l) yielded a comparable amount of aNSCs/NPCs (Supplementary Fig. 4j) but an increased amount of proliferating aNSCs/NPCs in $EGFL7^{-/-}$ mice (Fig. 4m–o). However, the amount of TAPs (Fig. 4p–r) and NBs (Fig. 4s–u) was reduced. The conditional EGFL7 knockout model $EGFL7^{fl/fl};Nestin\text{-}CreERT2$ ($EGFL7^{i\Delta NSC}$) phenocopied these observations and displayed a reduced amount of TAPs (Supplementary Fig. 4k) and NBs (Supplementary Fig. 4l), indicating that this effect was specific for adult NSCs. Data suggest ectopic expression of EGFL7 in the SVZ pushed aNSCs towards neuronal differentiation while the loss of EGFL7 delayed this process and caused an accumulation of proliferating aNSCs *in vivo*.

**EGFL7 changed the activation state of NSCs.** Initially, the amount of aNSCs *in vivo* was assessed by counting NICs; less spheres were recovered from AdEGFL7-infected SVZ tissue (Fig. 5a). Further, proliferating cells in the SVZ were labelled by

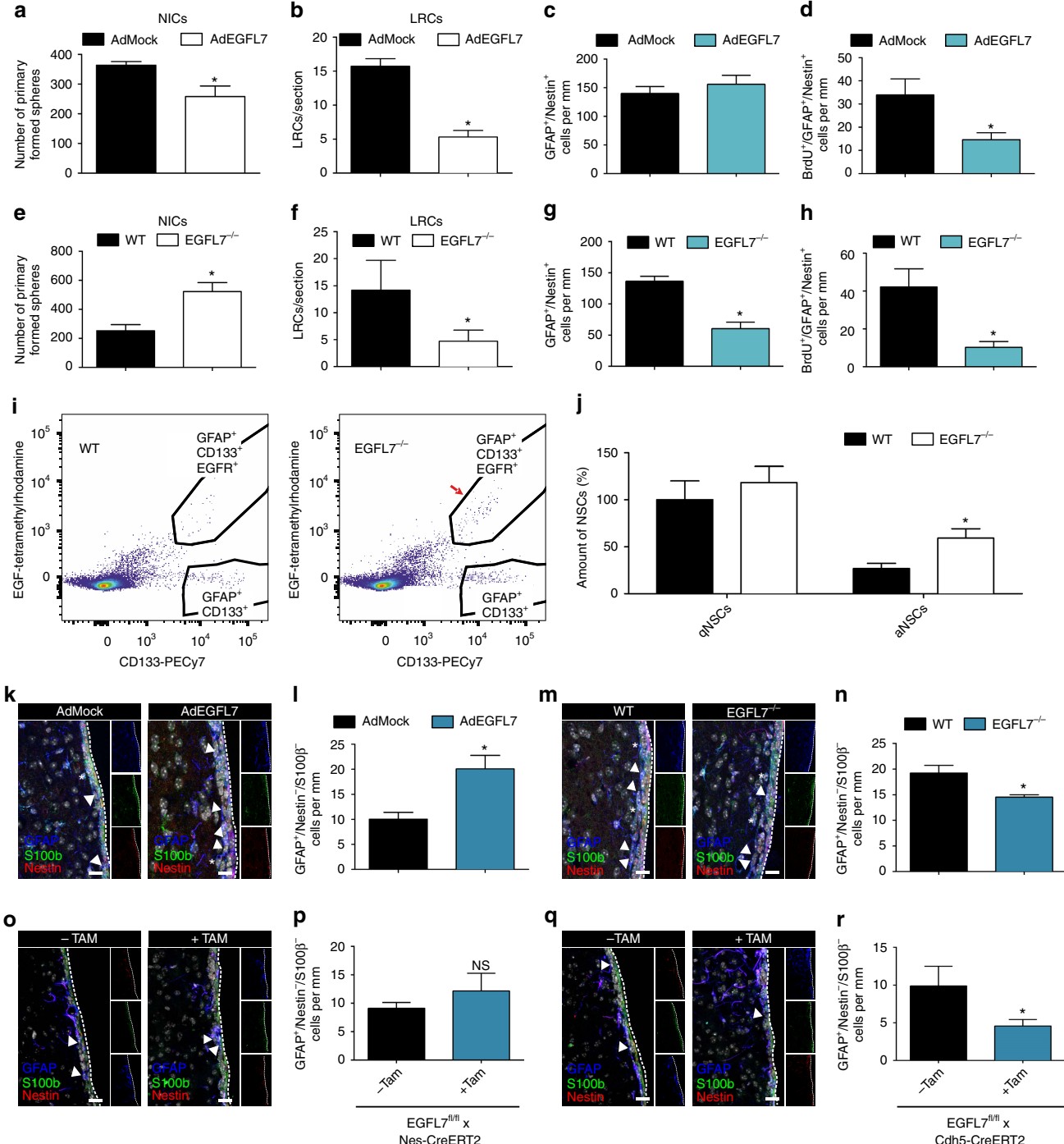

**Figure 5 | EGFL7 changed the activation state of NSCs.** (**a**) Ectopic EGFL7 *in vivo* resulted in less NICs (257.7 ± 35.66 versus 363.7 ± 12.47 spheres in AdMock; *n* = 3; *P* < 0.05) and (**b**) LRCs (15.73 ± 1.11 versus 5.31 ± 0.96 cells per section in AdMock; *n* = 6; *P* < 0.005). (**c**) AraC eliminated proliferating cells in the SVZ and accompanying ectopic EGFL7 did not affect total aNSCs/NPCs (156.2 ± 15.74 versus 140.1 ± 11.98 cells per mm in AdMock; *n* = 6; *P* < 0.01) but (**d**) reduced proliferating aNSCs/NPCs (14.69 ± 2.95 versus 33.89 ± 6.91 cells per mm in AdMock; *n* = 6; *P* < 0.05) 10 d post regeneration. (**e**) *EGFL7*$^{-/-}$ mice yielded more NICs (522.6 ± 21.49 versus 252.0 ± 14.85 spheres in WT; *n* = 8; *P* < 0.001) and fewer (**f**) LRCs (5.9 ± 1.22 versus 11.5 ± 1.28 cells per section in WT; *n* = 6; *P* < 0.05). (**g**) AraC treatment reduced aNSCs/NPCs by half (60.25 ± 10.5 versus 136.3 ± 7.93 cells per mm in AdMock; *n* = 6; *P* < 0.005) and thereby (**h**) reduced proliferating aNSCs/NPCs (10.25 ± 3.15 versus 42.25 ± 9.52 cells per mm in WT; *n* = 6; *P* < 0.05) 10 d post regeneration. (**i,j**) Quantification of SVZ-derived NSCs by FACS (markers GFAP, CD133, EGFR) revealed more aNSCs (59.09 ± 10.1 versus 26.82 ± 5.45% in WT; *n* = 6; *P* < 0.05) in the SVZ of *EGFL7*$^{-/-}$ animals. (**k–r**) IF analyses using the marker combinations GFAP$^+$/Nestin$^-$/S100β$^-$ (qNSCs, arrowheads) or GFAP$^+$/Nestin$^-$/S100β$^+$ (differentiated astrocytes, asterisks) revealed (**l**) more qNSCs (20.07 ± 2.72 versus 10.03 ± 1.36 cells per mm in AdMock; *n* = 3; *P* < 0.05) upon ectopic expression of EGFL7 but (**n**) a decrease in *EGFL7*$^{-/-}$ mice (14.52 ± 0.47 versus 19.24 ± 1.47 cells per mm in WT; *n* = 3; *P* < 0.05). (**o,p**) aNSC/NPC-specific loss of EGFL7 did not affect qNSCs *in vivo* while (**q,r**) the blood vessel-specific knockout did (9.88 ± 2.63 versus 4.57 ± 0.86 cells per mm in WT; *n* = 3; *P* < 0.05). Statistics performed by Student's *t*-test/Mann–Whitney *U*-test. Error bars indicate s.d. Scale bar, 20 µm.

injection of the base analog BrdU. The number of label-retaining cells (LRCs) in the SVZ was counted 4 weeks later as a measure for actively proliferating cells that had returned to quiescence. A reduced number of LRCs formed in the SVZ of AdEGFL7-infected mice (Fig. 5b).

Subsequently, the time course of cellular reconstitution of the SVZ on removal of all proliferating cells was assessed. To achieve this, the anti-mitotic drug AraC was infused into the lateral ventricle for 1 week using osmotic pumps and paralleled by CVI of AdEGFL7 or AdMock. Subsequently, BrdU was administered for 3 d to label the proliferating cells replenishing the stem-cell niche. The cytoarchitecture of the SVZ was analysed 10 d after disposal of AraC by IF (Supplementary Fig. 5a). AraC led to depletion of the SVZ from all proliferating cells, including aNSCs/NPCs, TAPs and NBs but did not affect slowly dividing cells such as qNSCs (Supplementary Fig. 5b,c). The total amount of aNSCs/NPCs was not significantly altered (Fig. 5c) but lower numbers of proliferating aNSCs/NPCs were counted in AdEGFL7-infected mice (Fig. 5d). This caused a decrease in newborn TAPs (Supplementary Fig. 5d), NBs (Supplementary Fig. 5e) and proliferating cells in total (Supplementary Fig. 5f). In conclusion, expression of EGFL7 delayed the repopulation of the NSC niche in the SVZ upon AraC treatment, indicating that ectopic EGFL7 pushed aNSCs towards quiescence.

Further, more NICs were recovered from the SVZ of $EGFL7^{-/-}$ mice (Fig. 5e) and the amount of LRCs in $EGFL7^{-/-}$ mice was decreased (Fig. 5f), indicating that less aNSCs returned to quiescence and/or were reactivated more rapidly. Application of AraC to $EGFL7^{-/-}$ mice, which harbour an increased aNSC pool, reduced the absolute amount of aNSCs/NPCs by about 50% (Fig. 5g). Consequently, a delayed repopulation of the SVZ with aNSCs/NPCs (Fig. 5h), TAPs (Supplementary Fig. 5g), NBs (Supplementary Fig. 5h) and proliferating cells (Supplementary Fig. 5i) was observed. Data indicate that the loss of EGFL7 caused an increase in aNSCs at the expense of qNSCs, which was quantified by FACS analysis[2]. A higher proportion of aNSCs was retrieved from the SVZ of $EGFL7^{-/-}$ mice (Fig. 5i,j), verifying that the loss of EGFL7 caused a shift from qNSCs into aNSCs. IF analysis of SVZ tissue by the marker combination GFAP$^+$/Nestin$^-$/S100β$^{-2}$ revealed an increased amount of qNSCs upon CVI of AdEGFL7 (Fig. 5k,l) but a decrease in $EGFL7^{-/-}$ qNSCs (Fig. 5m,n). The amount of differentiated GFAP$^+$/Nestin$^-$/S100β$^+$ astrocytes, however, was not significantly altered upon AdEGFL7 CVI or in $EGFL7^{-/-}$ animals (Supplementary Fig. 5j,k). Subsequently, we compared the amount of qNSCs in mice specifically lacking EGFL7 in aNSCs ($EGFL7^{fl/fl}$;Nestin-CreERT2) or blood vessels ($EGFL7^{fl/fl}$;Cdh5-CreERT2). While the former mice remained unaffected (Fig. 5o,p), the latter displayed a significant decrease in qNSCs (Fig. 5q,r). Data suggest that vascular rather than neural EGFL7 affected the activation state of NSCs in the SVZ.

**EGFL7 altered neurogenesis and behaviour.** The influence of EGFL7 on NSCs/NPCs suggested an effect on neurogenesis, therefore the amount of adult-born neurons in the OB was determined. Proliferating cells were labelled with BrdU (Fig. 6a) and 28 d later more adult-born neurons were counted in the OB of AdEGFL7-infected mice (Fig. 6b–d). Conversely, a reduced number of newborn cells was detected in the OB of $EGFL7^{-/-}$ (Fig. 6e–g) or tamoxifen-induced adult $EGFL7^{fl/fl}$;Nestin-CreERT2-knockout mice (Fig. 6h). However, the absolute sizes of the brains, ventricles, OBs and OB substructures, measured by magnet resonance imaging, were not changed in $EGFL7^{-/-}$ mice (Supplementary Fig. 6a).

Consequences for neuronal processing were analysed using extracellular electrophysiological recordings in the dorsal mitral cell layer of the mid-anterior OB (Supplementary Fig. 6b–h). Interestingly, the firing rate of mitral neurons in $EGFL7^{-/-}$ mice in response to the olfactory stimulant amyl acetate was increased (Fig. 6i). Cross-correlation analysis of simultaneously recorded neurons in $EGFL7^{-/-}$ and WT mice revealed a majority of synchronous interactions in WT as opposed to majorly asynchronous interactions in $EGFL7^{-/-}$ mice (Fig. 6j). Together, data suggest that the feedback inhibition accounting for the synchrony of mitral cells was affected in $EGFL7^{-/-}$ mice, resulting in an increased firing rate and diminished synchrony of cofiring neurons upon olfactory stimulation.

To determine whether this abnormal neuronal transmission in the OB affected olfactory sensing, behavioural assays were performed in $EGFL7^{-/-}$ mice. Animals were not impaired in their general activity, motor balance, sensorimotor gating, social behaviour, anxiety or pain (Supplementary Fig. 6i–n). However, $EGFL7^{-/-}$ mice displayed abnormalities in olfactory habituation/dishabituation tests. $EGFL7^{-/-}$ and WT mice displayed comparable exploration and habituation of the neutral scent water but WT mice displayed increased exploration of the scent vanilla, while $EGFL7^{-/-}$ did not (Fig. 6k). The same counts using urine of an unfamiliar male (Fig. 6l). Further, olfactory tests based on the innate mouse avoidance of particular scents were performed. Increasing concentrations of 2-methylbutyric acid (2-MB), a component of spoiled food, were presented in different concentrations to the animals. Comparing the exploration time of increasing 2-MB dilutions with water revealed increasing avoidance of this scent by WT but not $EGFL7^{-/-}$ animals (Fig. 6m). Even more striking was a test using trimethylthiazoline (TMT), a constituent of fox urine. Essentially, WT littermates sensed TMT at each dilution and never explored it like water. However, $EGFL7^{-/-}$ mice reacted to TMT only at very high concentrations (Fig. 6n). In the absence of any gross anatomical abnormalities of the olfactory epithelium (Supplementary Fig. 6o–q) or behavioural peculiarities of $EGFL7^{-/-}$ mice it can be concluded that these animals were impaired in their ability to sense odours.

## Discussion

The NSC niche in the SVZ maintains radial glia-derived NSCs which undergo a fascinating metamorphism upon activation, which eventually allows them to become functional neurons in the OB[1]. NSCs remain in an undifferentiated, quiescent state and need to enter the cell cycle before being able to differentiate into neuronal progenitors and mature neurons. Many of these processes take place in the close proximity of blood vessels within a specialized vascular niche and are indeed regulated by blood vessel-derived proteins[4,9,10]. However, only a few of these vascular factors have been described. In the current work, the potential of EGFL7 to act as a factor shaping the microenvironment in the vascular niche to regulate NSCs and neurogenesis in the SVZ was analysed.

Previously, EGFL7 has been described to govern blood vessels (reviewed by Nikolic et al.[28]) and as a regulator of NSCs in vitro[22]. Our developmental studies identified two waves of EGFL7 expression in the murine embryo. One around E9 when much of the vascular network is formed and when EGFL7 expression was restricted to ECs. The other started in neurons around E18 and reached its maximum at P40. Previous developmental studies focused on the vascular functions of EGFL7 and the early expression patterns at E9.5-E13.5, leaving the late onset expression of EGFL7 in neurons undiscovered[29].

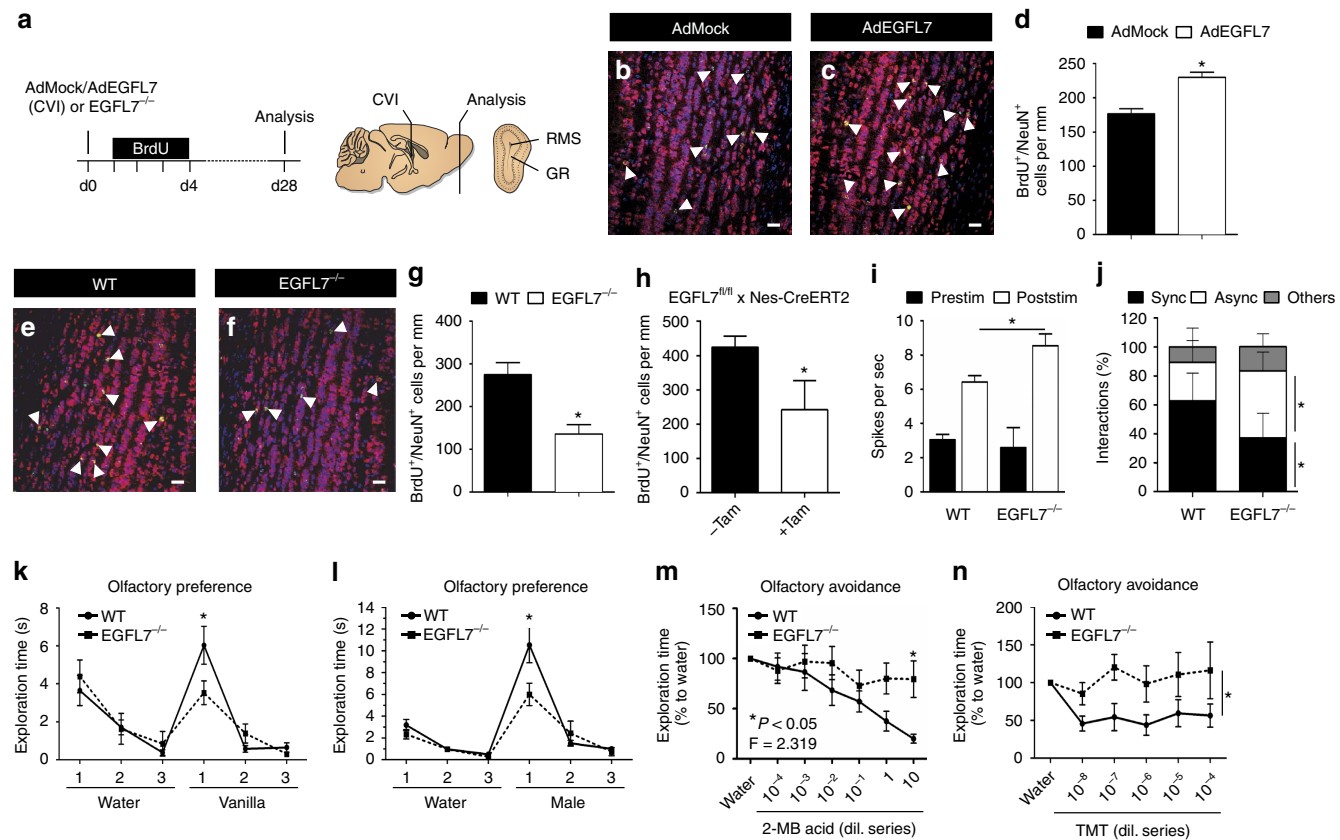

**Figure 6 | EGFL7 altered neurogenesis and behaviour. (a)** Experimental set-up. **(b–d)** Ectopic EGFL7 increased the amount of adult-born neurons (IF; $229.8 \pm 7.56$ versus $176.7 \pm 7.14$ cells per mm in AdMock; $n = 6$; $P < 0.005$), while **(e–g)** less neurons were detected in the OB of $EGFL7^{-/-}$ ($135.9 \pm 21.79$ versus $274.6 \pm 28.35$ cells per mm in WT; $n = 6$; $P < 0.005$) or **(h)** $EGFL7^{i\Delta NSC}$ mice ($242.0 \pm 42.37$ versus $424.3 \pm 18.68$ cells per mm in WT; $n = 3$; $P < 0.05$). **(i,j)** Electrophysiological conduction from mitral cells in the OB of $EGFL7^{-/-}$ mice revealed **(i)** an increase in the average evoked firing rate of $EGFL7^{-/-}$ mice upon amyl acetate exposure ($8.55 \pm 1.17$ versus $6.41 \pm 0.66\%$ in WT; $n(WT) = 53$, $n(EGFL7^{-/-}) = 43$; $P < 0.05$). **(j)** Comparison of the proportions of synchronous (black boxes) and asynchronous firing neurons (white boxes) or other interacting neurons (grey boxes) upon exposure to amyl acetate revealed a majority of asynchronous firing ($46 \pm 13$ versus $26 \pm 14\%$ in WT; $P < 0.05$; $n(WT) = 53$, $n(EGFL7^{-/-}) = 43$) in $EGFL7^{-/-}$ mice but synchronous firing in WT ($37 \pm 15$ versus $63 \pm 19\%$ in WT; $P < 0.05$; $n(WT) = 53$, $n(EGFL7^{-/-}) = 43$). **(k,l)** $EGFL7^{-/-}$ mice displayed a deficit in olfactory preference for both **(k)** artificial (water/vanilla; $3.53 \pm 2.16$ versus $6.03 \pm 4.23$ s in WT; $n = 12$; $F_{(2, 56)} = 4.047$, $P < 0.05$, two-way ANOVA for repeated measures) and **(l)** natural (water/male urine; $6.0 \pm 3.42$ versus $10.53 \pm 7.35$ s in WT; $n = 12$; $F_{(2, 50)} = 5.891$, $P = 0.005$, two-way ANOVA for repeated measures) scents. **(m,n)** $EGFL7^{-/-}$ mice displayed elevated tolerance to **(m)** 2-MB ($79.49 \pm 81.3$ versus $20.1 \pm 23.73\%$ in WT; $n = 20$; $F_{(6, 282)} = 2.319$, $P < 0.05$, two-way ANOVA for repeated measures) and **(n)** TMT ($116.36 \pm 99.25$ versus $56.36 \pm 55.13\%$ in WT; $n = 13$; $F_{(1, 18)} = 7.351$, $P < 0.05$, two-way ANOVA for repeated measures) in olfactory avoidance tests. Data suggest loss of EGFL7 impaired olfactory perception. Scale bar, 50 μm **(b,c,e,f)**.

In the SVZ EGFL7 was readily detectable[22], which offered the possibility that ChAT[+] neurons, themselves regulating neurogenesis in the SVZ[30], transported EGFL7 there by their axons. However, expression studies revealed that neurons are only a minor source of EGFL7 in the SVZ, but ECs and NSCs/NPCs synthesize the majority of EGFL7. NSCs reside in close proximity to blood vessels[15], putatively, regulated by hypoxia, which regulates both, EGFL7 expression[31] and neurogenesis[32].

$EGFL7^{-/-}$ neurospheres displayed an increased self-renewal potential and became larger, which was likely mediated by the previously reported inhibition of Jagged1-induced Notch1 signalling[22]. Comparison of the transcriptomes of $EGFL7^{-/-}$ and WT neurospheres revealed that $EGFL7^{-/-}$ spheres experienced a delay in differentiation *in vitro*. WT spheres upregulated proteins of neuronal signalling at d5, for example, glutamate receptors or synaptic proteins, whereas $EGFL7^{-/-}$ spheres retained a protein expression profile similar to d2 and indicative of increased metabolic activity[2]. We hypothesize that delayed differentiation and sustained proliferation of $EGFL7^{-/-}$ aNSCs caused larger neurospheres.

However, the influence EGFL7 exerted on Notch signalling as measured by Hes5 levels in neurospheres and SVZ tissue was, in part, in opposition. A comparable discrepancy has been described for the canonical Notch ligands Jagged1 and Dll1, which increased the self-renewal potential of neurospheres *in vitro*[33,34] but kept NSCs quiescent *in vivo*[18,35]. Data obtained in neurospheres may therefore not directly be transferred to NSCs in the SVZ. This could be due to the focus of this assay on aNSCs and their progeny but also due to a lack of a proper stem-cell niche[36]. We hypothesized that EGFL7 may exerted its effect on Hes5 by Notch signalling components differentially expressed in SVZ tissue and neurospheres. An expression screen of various Notch signalling components identified the Notch ligand Dll4 as a potential niche component that was expressed on blood vessels but absent from neurospheres.

The canonical Notch ligands Dll4 and Jagged1 interact with Notch1 in two different ways. Either paracrine *in trans*, which activates Notch signalling, or autocrine *in cis*, which inhibits Notch[37]. Molecular interaction analyses displayed preferential binding of EGFL7 to the region in Notch1 that is responsible for

activation in trans (EGF-like repeats 11 and 12) but not the ones engaged for Notch deactivation in cis (EGF-like repeats 24 to 29). Therefore it could be concluded that EGFL7 predominantly affected Notch1 activation in trans. In the neurovascular stem-cell niche the interaction between Dll4 and Notch1 may only occur in trans because Dll4 expression is restricted to blood vessels and is therefore solely activating. Jagged1, however, is ubiquitously expressed[18,20,33] and may either activate or inhibit Notch signalling in NSCs. Furthermore, EGFL7 competed with Jagged1 for binding to Notch1, while it cooperated with Dll4, probably due to the direct binding of EGFL7 and Dll4[22]. Previously, this ligand has been discussed in the context of NPCs[18,38] but an effect on NSCs has not yet been described. These observations allowed for the hypothesis that elevated levels of EGFL7 in the SVZ in vivo inhibited Jagged1 binding to Notch1 in trans but this loss was compensated by the increased recruitment of Dll4. However, in neurospheres in vitro, Dll4 was not present and no compensation for Jagged1 could occur in the presence of high levels of EGFL7. Therefore, only the Notch inhibiting potential of Jagged1 was executed under this condition. In conclusion, we suggest that the opposing effects EGFL7 exerts on Notch signalling in vitro and in vivo resulted from a combination of its preferred binding site in Notch1 and its differential influence on Dll4 and Jagged1 binding to Notch1. Furthermore, we could show that blood vessel-derived Dll4 pushed NSCs towards quiescence in vivo. Assuming EGFL7 affected NSCs in part by vascular Dll4, it was to be expected that EGFL7 pushes aNSCs towards quiescence as well. Conversely, an accumulation of proliferating aNSCs in EGFL7$^{-/-}$ mice was expected in vivo.

Indeed, ectopic expression of EGFL7 pushed aNSCs towards differentiation into TAPs, NBs and eventually interneurons. This was probably due to the transient inhibition of Jagged1-mediated Notch signalling in aNSCs and therefore comparable to the situation in RBPJ$^{-/-}$ mice[19], where the induced knockout of this central Notch signalling component led to a transient activation of neurogenesis in the adult SVZ. Conversely, loss of EGFL7 caused an accumulation of aNSCs within the SVZ and a smaller proportion of them entered the neuronal differentiation pathway. Consequently, less interneurons were formed in the OB of adult EGFL7$^{-/-}$ mice. This indicates that EGFL7 is necessary for neuronal differentiation of NSCs but also affects their activity state. Quantitative comparison of both pools in the SVZ by FACS[2,39,40] revealed that EGFL7$^{-/-}$ mice harboured an enlarged pool of aNSCs, which was verified by primary explant assays (NICs)[2,41].

AraC-mediated depletion of mitotic cells in the SVZ of EGFL7$^{-/-}$ mice reduced the total amount of aNSCs, indicating that a larger proportion of aNSCs were in an activated mitotic state at the time of the experiment as compared to WT. This effect was also reflected in the amount of LRCs in the SVZ. Actively proliferating cells were labelled with BrdU. Subsequently, cells either departed towards the OB or returned to quiescence to form LRCs, which were markedly reduced in EGFL7$^{-/-}$ mice after 1 month. However, two scenarios are possible and either less aNSCs returned to quiescence in the first place or the ones that did were reactivated more rapidly. In both scenarios EGFL7 was needed to keep NSCs quiescent. Live microscopy on NSCs in vitro[42] revealed that EGFL7$^{-/-}$ aNSCs differentiated but after a period of rest, re-entered cell cycle. Apparently, EGFL7 is needed to keep NSCs quiescent.

To define whether or not the regulation of NSCs/NPCs by EGFL7 is of physiological relevance for the adult brain, the amount of adult-born neurons in the OB was determined and was found to be reduced in EGFL7$^{-/-}$ but increased in AdEGFL7-infected mice. Mitral and tufted cells, the two main excitatory

neural cells in the OB, form embryonically[43], while the majority of interneurons is generated postnatally[44] and was found altered by EGFL7. This life-long modification of the local circuit in the OB with interneurons alters olfactory information processing and is the basis for olfactory learning[45,46]. Neuronal conductance of mitral cells in the OB of EGFL7$^{-/-}$ mice showed a higher signal departing towards the telencephalon, which was due to a reduction in the amount of inhibitory neurons. However, the loss of inhibitory interneurons occurred at the expense of synchronization of mitral cell activity[47]. This increased the threshold for odorant detection in olfactory behaviours and reduced olfactory discrimination, which is in agreement with previous studies on the impact of mitral cell activity on mouse behaviour and the role of adult-born neurons in this context[48].

Collectively, this work identifies EGFL7 as a neurovascular regulator of NSCs, governing olfactory perception and behaviour. On the molecular level, EGFL7 supported vascular Dll4 in pushing aNSCs towards quiescence but inhibited neural Jagged1 to prevent an excessive expansion of the activated stem cell pool. Future work will reveal whether the downregulation of EGFL7 and similar factors may be applied to reconstitute or at least increase the aNSC pool in aged animals to delay the cognitive decline happening during aging.

## Methods

**Mouse models.** Constitutive EGFL7$^{-/-}$ mice were generated and kindly provided by Weilan Ye, Genentech[49]. The knockout was achieved by insertion of a retroviral gene trap vector upstream of intron 2 of the egfl7 gene. The insertion leads to silencing of endogenous Egfl7 transcription, and transcripts initiated from the inserted vector contain stop codons in all three frames, thus abolishing EGFL7 protein synthesis. Conditional EGFL7$^{fl/fl}$ mice were created by insertion of one loxP cassette upstream of exon 3 and a second one downstream of exon 7, to remove all putative start codons from the egfl7 gene upon Cre-mediated recombination (ingenious targeting laboratory, Ronkonkoma, USA). Dll4$^{fl/fl}$ mice were provided by Rui Benedito (CICD, Madrid, Spain), Cdh5(PAC)-CreERT2 mice originated from Ralf Adams' lab (Max Planck Institute for Molecular Biomedicine, Münster, Germany). Nestin-CreERT2 mice were provided by Beat Lutz (Institute of Physiological Chemistry, Mainz, Germany) and originated from Günther Schütz's lab (DKFZ, Heidelberg, Germany)[50]. In general, mice were in a C57BL/6J background and have been back-crossed for at least six generations. Two-month-old male Dll4$^{fl/fl}$;Cdh5(PAC)-CreERT2, Dll4$^{fl/fl}$;Nestin-CreERT2 or EGFL7$^{fl/fl}$;Nestin-CreERT2 mice (n = 6 each) received repetitive daily injections of tamoxifen in peanut oil/ethanol (9:1) as a vehicle at a dose of 0.15 mg kg$^{-1}$ body weight for five consecutive days to create adult Dll4$^{iΔEC}$ mice, lacking Dll4 in blood vessels or Dll4$^{iΔNSC}$ and EGFL7$^{iΔNSC}$ mice, lacking the respective protein in stem cells and their progeny. Animals equally treated with vehicle only served as negative controls. Animal experiments were approved by the ethics committee of the Landesuntersuchungsamt Rheinland-Pfalz, and were performed according to German Federal Law §8 Abs. 1 TierSchG.

**Cerebroventricular injections.** Two-month-old male C57BL/6J mice (n = 20) were anaesthetised by intraperitoneal (i.p.) administration of ketamine (100 mg kg$^{-1}$ body weight) and xylazine (16 mg kg$^{-1}$ body weight) in 0.9% sodium chloride (NaCl) solution. To avoid hypothermia, mice were kept on an electric blanket during the entire duration of surgery. In addition, ointment (Bepanthen, Bayer, Leverkusen, Germany) was applied on the eyes to prevent their dehydration. While the animals were under deep anaesthesia, a 0.5 cm longitudinal incision was made with a scalpel above the position of the bregma, which was subsequently visualized by cleaning the calvarium with 30% H$_2$O$_2$ solution. Mice were fixed in a stereotaxic frame (Kopf Instruments, Tujunga, CA, USA) and 5 µl of either EGFL7 expressing or control adenovirus (5 × 10$^6$ particles per µl) were injected during 2 min at the site 0.7 mm lateral and 2 mm ventral to the dura. The needle was slowly removed after 2 min delay for ventricular pressure equalization. Subsequent to clipping the cut, the wound was treated with xylocain to locally anaesthetise this area for several hours. For long-term experiments, mice received BrdU injection for 3 d starting from post-operative day 1. At the post-operative d 4 (short-term) or 28 (long-term), mice were killed by transcranial perfusion with 4% paraformaldehyde in PBS. Brains were removed, fixed in 4% paraformaldehyde in PBS at 4 °C for 24 h and subsequently transferred into 30% sucrose in PBS for cryo-protection. Brains were embedded in Tissue-Tek O.C.T. (Sakura Fintek, Torrance, CA, USA) compound, frozen on dry-ice and stored at −80 °C until further processing. Coronal cryosections were prepared at a thickness of 10 µm and stored at −20 °C.

**AraC treatment.** Brain infusion kits combined with Mini-osmotic pumps (2001, Alzet, Cupertino, CA, USA) were filled with 2% AraC (Sigma, St Louis, MI, USA) and implanted on the surface of the brain as described[1,8,10]. Pumps were implanted either subsequent to virus injection in the case of EGFL7 gain-of-function experiments or without prior injection in the case of EGFL7 loss-of-function experiments using identical stereotactic coordinates. Six days after AraC-infusion, mice received BrdU injections for 3 d. Mice were killed after 10 d of regeneration by transcardial perfusion.

**Electrophysiology.** Extracellular recordings were carried out on three WT and three $EGFL7^{-/-}$ mice 3–5 months in age. Mice were deeply anaesthetised with i.p. injection of a mixture of ketamine (stock 100 mg ml$^{-1}$ Ketanarkon, Streuli Pharma AG, Uznach, Switzerland) and xylazine (stock 20 mg ml$^{-1}$ Xylazine, Streuli Pharma AG, Uznach, Switzerland) at a final concentration of 100 mg kg$^{-1}$ and 10 mg kg$^{-1}$ in sterile saline solution (NaCl 0.9%, Braun Medical SA, Sempach, Switzerland), respectively. The withdrawal reflex was checked every 30 min to monitor anaesthesia. Body temperature was maintained at 37.5 °C using a heating pad connected to a temperature sensor placed intrarectally (Lis-Medical, Frankfurt, Germany). The mouse was mounted into a stereotaxic frame for surgery. The surface of the skull above the OB was cleaned and an incision was made to open a small window on top of the OB. Single glass coated, platinum- and gold-plated electrodes with low impedance (1–2 MΩ) were used for extracellular recordings[51]. The electrodes were advanced at a 45° angle from the right side into the dorsal OB to reach the mitral cell layer using a micromanipulator (Kopf Instruments) at the coordinates: AP = +4.05 to 4.95 mm from bregma, ML = 1.2–1.5 mm and DV = 200–600 μm. After the first penetration the surface of the OB was protected using 3% pure agarose gel (Carl Roth, Karlsruhe, Germany). An average of seven electrode penetrations was performed in the mid-anterior OB per animal. The signal from the electrode was amplified and filtered (400–2,000 Hz), visualized on an oscilloscope, and conveyed to an amplifier Power 1401 (Cambridge Electronic Design Limited, Cambridge, UK). The signal was digitized (10,000 Hz sampling rate) using Spike 2 (Cambridge Electronic Design Limited, Cambridge, UK). The signals were recorded for ~1.5 min of spontaneous activity followed by 1 s of odour stimulation and 30 s of evoked activity. Concentration of 4.4 μM of an amyl acetate dilution was presented to the animal using an olfactometer[52]. The 1 s odour exposure was recorded as a triggered signal. Following the recording session electrolytic lesions were induced at specific depths using 3 pulses of 5 mA for 7 s at an interval of 5 s. The recorded waveforms were analysed post hoc from electrode penetration sites in the mitral cell layer only. Spike units corresponding to one cell were sorted from each recording using Spike 2 (Cambridge Electronic Design Limited, Cambridge, UK). Peristimulus time histograms for amyl acetate were calculated for all sorted spike traces using the Neuroexplorer software (Nex Technologies, Madison, Alabama, USA). The mean spontaneous (prestim) and evoked (poststim) firing rate for each cell was obtained from the auto-renewal density plot using the Neuroexplorer software. Cross-correlation analysis was performed between each cell pair in the multicellular recordings using the Neuroexplorer software. Cross-correlograms characterized by a symmetric hump near lag 0 indicated that the cells fire synchronously. Cross-correlograms with no hump or asymmetric hump near lag 0 were characterized as non-synchronous. The distribution between cell pairs firing synchronously and asynchronously, in presence of amyl acetate in WT and $EGFL7^{-/-}$ was analysed by two-way Anova. Firing rates and cumulative frequencies and correlations of mitral cells were carried out using non-directional Student's t-test or Mann–Whitney U-test. The peristimulus responses were analysed by one-way Anova with two factors.

**Olfactory habituation/dishabituation behavioural test.** The test was performed as described[53] with minor modifications using adult littermates. This test consists of sequential presentations of two different odours: water and vanilla. In another experimental session, water and urine samples of an unfamiliar male mouse were used. Each odour (or water) is presented in three consecutive trials of 2 min per trial with 1 min intertrial interval. Water solutions of the food odourants were used. Each mouse was presented with fresh odour sample (1 μl) on filter paper. Odour samples were presented such that mice have no opportunity to touch/bite the sample (that is, enclosed in small perforated Petri dish). Presentations were performed in the clean standard mouse cage. Before presentations animals were habituated for 30 min in this cage. Cumulative sniffing time was manually recorded during each presentation of odour. Sniffing is scored when the animal is orienting towards the dish with its nose 2 cm or closer to the dish.

**Olfactory sensitivity behavioural test.** Olfactory sensitivity was behaviourally evaluated by measuring an innate olfactory avoidance behaviour for presentation of spoiled foods (2-methylbutyric acid, 2-MB) and predators (trimethylthiazoline, TMT) scents, based on previously described protocol[53]. Multiple dilutions (reported in the figures) of the scent were tested in an increasing concentration fashion, that is, initially water, followed by the lowest concentration of the scent. To minimize a potential confounding effect of multiple testing, a test session was performed with 24 h intervals. The test cage (20 × 15 × 13 cm) was virtually divided into two equal zones, so mice could move freely between the zones. Animals were first habituated to the cage for 10 min, and then a filter paper with 10 μl of test odourant was introduced into one of the zones. Time (in s) spent in each zone was measured by Ethovision video-tracking system (Noldus Inc., Wageningen, Netherlands) during the 3 min test period. Avoidance was calculated in per cent, taking as 100% the time spent in zone with water sample. Responses to 2-MB and TMT were measured sequentially (2-MB first). A freshly washed and autoclaved test cage were used for each test session.

**Elevated plus maze behavioural test.** The animal was placed in the central platform facing an open arm of the plus maze (made of grey Perspex with a central 5 × 5 cm central platform, 2 open arms, 30 × 5 cm and 2 enclosed arms, 30 × 5 × 15 cm, illumination 300 lux). Behaviour was recorded for 5 min by an overhead video camera and a personal computer (PC) equipped with Ethovision (Noldus Inc.) software to calculate the time each animal spent in open or closed arms. The proportion of time spent in open arms was used for estimation of open arm aversion (fear equivalent).

**Open field behavioural test.** Spontaneous activity in open field was tested in a grey Perspex arena (120 cm in diameter, 25 cm high). The animal was placed in the centre of the open field and was allowed to explore it for 7 min. The behaviour was recorded by a PC-linked overhead video camera. Ethovision (Noldus Inc.) software was used to calculate the distance travelled and the time spent in the central versus the intermediate and peripheral zones of the open field.

**Rotarod test.** Rotarod is a test for motor function, balance and coordination and comprises a rotating drum, which is accelerated from 4 to 40 revolutions per min over the course of 5 min. Mice were placed individually on the revolving drum (Ugo Basile, Comerio, Italy). Once they were balanced, the drum was accelerated. The time in seconds at which each animal fell from the drum was recorded using a stop-watch. Each animal received three consecutive trials, the longest time on the drum was used for analysis.

**Social behaviour test.** Social behaviour was evaluated in a sociability test as described[54]. The social testing apparatus was a rectangular, three-chambered box. Each chamber was 20 × 40 × 22 cm. Dividing walls were made from clear Plexiglas, with rectangle openings (35 × 35 mm) allowing access into each chamber. The chambers of the social apparatus were cleaned and fresh paper chip bedding was added between trials. The test mouse was first placed in the middle chamber and allowed to explore for 5 min. The doorways into the two side chambers were obstructed by plastic boxes during this habituation phase. After the habituation period, an unfamiliar C57BL/6J male (stranger), that had no prior contact with the subject mice, was placed in one of the side chambers. The location of stranger in the left versus right side chamber was systematically alternated between trials. The stranger mouse was enclosed in a small (60 × 60 × 100 mm), rectangle wire cage, which allowed nose contact between the bars, but prevented fighting. A weighted cup was placed on the top of the cage to prevent climbing by the test mice. The animals serving as strangers had previously been habituated to placement in the small cage. Both doors to the side chambers were then unblocked and the subject was allowed to explore the entire social test box for a 10 min session. The amount of time spent in each chamber was recorded by video-tracking system Ethovision (Noldus Inc.).

**Prepulse inhibition of acoustic startle response test.** To measure the startle reactivity, mice were placed in small metal cages (90 × 40 × 40 mm) which restrict major movements and exploratory behaviour. The cages are equipped with a movable platform floor attached to a sensor recording vertical movements of the floor. The cages are placed in four sound-attenuating isolation cabinets (Med Associates, St Albans, VT, USA). A startle reflex is evoked by acoustic stimuli delivered from a loudspeaker suspended above the cage and connected to an acoustic generator. The startle reaction of a mouse to the acoustic stimuli evokes a movement of the platform. The transient force resulting from this movement of the platform was recorded on a PC during a recording window of 260 ms and stored in the computer for further evaluation. The recording window is measured from the onset of the acoustic stimuli. An experimental session consisted of a 2 min habi-tuation to the 65 dB background white noise (continuous throughout the session), followed by a baseline recording for 1 min at background noise. After baseline recording, six pulse alone trials using the startle stimuli of 120 dB intensity and 40 ms duration were applied to decrease influence of within-session habituation. These data were not included in the analysis of the prepulse inhibition. For tests of prepulse inhibition, the 120 dB/40 ms startle pulse was applied either alone or preceded by a prepulse stimulus of 70, 75 or 80 dB intensity and 20 ms duration. An interval of 100 ms with background white noise was employed between each prepulse and pulse stimulus. The trials were presented in a pseudorandom order with an interval ranging from 8 to 22 s. Amplitude of the startle response (expressed in arbitrary units) was defined as a difference between the maximum force detected during a recording window and the force measured immediately before the stimulus onset. Amplitudes were averaged for each individual animal, separately for both types of trials (stimulus alone, stimulus preceded by a prepulse). Prepulse inhibition was calculated as a percentage of the startle response using the

formula: % prepulse inhibition = 100 − [(startle amplitude after prepulse – pulse pair)/(startle amplitude after pulse only) × 100].

**Hot plate test.** A standard hot plate (Ugo Basile) was used for the assessment of nociceptive sensitivity, widely used to measure thermal pain threshold. The plate was heated to 52 °C and the mouse was confined there by Plexiglas cylinder (diameter 19 cm, height 26 cm). Time (in seconds) to show hind paw response (licking or shaking) or jumping was noted as the pain threshold.

**Magnetic resonance imaging (MRI).** Brains of $EGFL7^{-/-}$ and control mice ($n = 13$ each) were fixed in formalin. For the MR examination, brains were processed in agarose (4%) to prevent movement within the scanner. The samples were placed in a mouse whole body coil (Bruker, Ettlingen, Germany) of a dedicated small animal ultra-high field MR scanner (ClinScan 7 Tesla, Bruker, Ettlingen, Germany). Imaging sequences and parameters were chosen as follows: T2-weighted imaging (slices 72, TR 2,000 ms, TE 46 ms, averages 2, voxel size $0.059 \times 0.059 \times 0.500$ mm, acquisition time 6:34 h). During data post-processing, the volumes of the brains, OB, granular cell layers as well as lateral, third and fourth ventricles were determined in mm³ using Osirix software and a dedicated plug-in for segmentation purposes (Chimaera, Erlangen, Germany).

**Olfactory epithelium.** Quantification of the olfactory epithelium within the nasal cavity of $EGFL7^{-/-}$ and WT mice was performed as previously described[55]. In brief, serial sections were analysed subsequent to HE-staining and were imaged using a BX51 microscope (Olympus, Tokyo, Japan) equipped with a Retiga 2000R CCD camera (Q-Imaging, Surrey, BC, Canada). Quantifications were performed using ImageJ software.

**Neurosphere assays.** Primary neurosphere cultures were prepared from adult C57BL/6J mice (8–12 weeks old) as previously described[56]. Mice were killed by cerebral dislocation. Brains were removed and the forebrain was opened under sterile conditions to access the SVZ for dissection. The tissue was enzymatically dissociated by incubation in Leibowitz L-15 medium, containing 0.8 mg ml⁻¹ Papain and 0.5 mM EDTA (dissociation medium, preactivated at 37 °C for 30 min) at 37 °C for 30 min. After centrifugation for 3 min at 300 g, the supernatant was removed, and the tissue was resuspended in ovomucoid inhibitor (Worthington, Lakewood, CO, USA). Cells were triturated with a fire-polished Pasteur pipette, collected by centrifugation for 3 min at 300 g, resuspended in DMEM/F-12 medium containing 1 mM HEPES, 10 µg ml⁻¹ penicillin, 10 µg ml⁻¹ streptomycin, 1 × B27 supplement and 20 ng ml⁻¹ EGF and FGF (Peprotech, Rocky Hill, CT, USA) (NSC medium) and plated in T75 flasks. Cells were incubated at 37 °C and 5% $CO_2$ and passaged every 5 d by enzymatic dissociation with accutase (PAA, Egelsbach, Germany) at 37 °C for 10 min under vigorous shaking.

NSC and progenitor populations were isolated from SVZ tissue by a BD FACS Aria II device using a 100 µm nozzle aperture and the marker combinations GLAST⁺/CD133⁺/EGFR⁻ (qNSCs), GLAST⁺/CD133⁺/EGFR⁺ (aNSCs), CD133⁻/EGFR⁺ (TAPs) or CD24⁺ (NBs). Cells were FACS sorted into 500 µl NSC medium in 24-well plates at a density of 500 cells per well. Neurospheres started forming on day 3.

For overexpression analyses, intact neurospheres (5 d in vitro) were enzymatically dissociated as described above. For each condition, 50,000 cells were plated in 6-well plates. Cells were treated with adenovirus (50 particles per cell) and grown for 5–6 d in NSC medium. Primary infected or $EGFL7^{-/-}$ spheres were dissociated, plated at clonal density (1 cell per µl) and grown for another 5–7 d. Alternatively, self-renewal assays were performed under clonal condition upon FACS sort (1 cell per well). In brief, neurospheres were dissociated and single cells were sorted into individual 96-well round-bottom wells (10 plates per condition) based on size and granularity (using the forward and the side scatter). Cells were cultured in standard medium supplemented with conditioned medium (ratio 1:1) and were grown for another 10 d. Subsequently, the number of secondary formed spheres as well as their diameter was determined using an IX70 microscope (Olympus, Tokyo, Japan). Statistically significant differences between groups were calculated using Student's $t$-test or Mann–Whitney $U$-test.

Cell cycle analysis was performed using Hoechst 33342 (Sigma-Aldrich)[57]. SVZ NSCs were incubated with 10 µg ml⁻¹ Hoechst 33342 for 45 min at 37° in culture medium. Hoechst 33342 was excited using ultraviolet laser. Single cells were gated using width and area parameters. The area parameter histogram was used to determine the percentage of cells in G1, S and G2M phases. Mann–Whitney $U$-test was used for statistical analysis.

**Time-lapse NSC/NPC microscopy.** Adult SVZ-derived NSCs were isolated and cultured as described[26,58,59]. In brief, time-lapse video microscopy and single-cell tracking of adult NSC/NPC cultures were performed using a cell observer (Zeiss, Oberkochen, Germany) and a TE-2000 microscope (Nikon, Tokyo, Japan), set at a constant temperature of 37 °C and 8% $CO_2$. Phase contrast/bright field images were acquired every 5 min for 5–7 d using a × 20 phase contrast objective (Zeiss/Nikon), an AxioCam HRm camera (Hamamatsu) and AxioVision 4.7/Metamorph software (Zeiss). For subsequent single-cell tracking a self-written

computer program (TTT)[60] was employed. Movies were assembled using Image J 1.42q (National Institute of Health, USA) software and are played at a speed of 1–3 frames per s. Under these culture conditions, WT aNSCs/NPCs exhibited a prototypical differentiation pattern and yielded a prototypical lineage tree as usually observed by single-cell tracking[26,42,61]. Briefly, slowly dividing astroglia (aNSCs) were characterized by a slow cell cycle, significant cell growth before division and the capacity to give rise to asymmetric lineage trees. Slowly dividing astroglia gave rise to fast dividing astroglia (activated astroglia) that in turn generated either TAPs or quiescent astroglial cells. Finally, TAPs gave rise to NBs after a limited number of divisions (Supplementary Movie 1). Characteristics described in vitro for aNSCs lineage progression were confirmed by BrdU-chasing or multicolour based clonal analysis in vivo[62,63]. Importantly, once the quiescent astroglial morphology had been acquired, WT cells remained quiescent until the end of the time-lapse imaging experiment, unless cells were challenged with mitogen growth factors that induced re-entry into the cell cycle[26].

**RNA isolation and reverse transcription.** Mouse embryos or postnatal brains were mechanically disrupted using mortar and pestle precooled in liquid nitrogen. Disruption of NSC and mouse lung endothelial cells (EC) was achieved by using Trizol Reagent (Invitrogen, Carlsbad, CA, USA) as suggested by the manufacturer. Disrupted tissues and cells were further homogenized by QIAshredder columns (Qiagen, Hilden, Germany) and RNA was purified using RNeasy mini kit (Qiagen) according to manufacturer's instructions. Concentration and purity of RNA were determined using the Bioanalyzer 2000 automated electrophoresis system (Agilent, Santa Clara, CA, USA). An amount of 1 µg of total RNA per sample was used to synthesize cDNA using oligo-d(T) primers and avian reverse transcriptase (iScript cDNA Synthesis Kit, Bio-rad, Hercules, CA, USA) according to manufacturer's protocol. Concentration and purity of generated cDNA were determined photometrically by absorption at 260 and 280 nm wavelengths (Biophotometer, Eppendorf, Hamburg, Germany).

**Quantitative real-time PCR (qRT-PCR).** qRT-PCR was performed using SYBR Green fluorescence mix (Thermo Fisher Scientific, Waltham, USA). Template cDNA weighing 130 ng and 1 pmol of gene-specific primers were used per reaction. PCR reaction and amplicon detection were performed by the iCycler real-time PCR system (CFX Connect RT-PCR System, Bio-Rad, Hercules, CA, USA). Quantification of expression was normalized according to the relative levels of cDNA in the samples based on quantitative analysis of the housekeeping genes (HKG) glyceraldehyde-3-phosphate dehydrogenase , RNA polymerase II (RNAPII) and ribosomal protein S13 (Rps13). Values were normalized using the CFX Manager Optical System Software (Bio-Rad). See Supplementary Table 1 for details on primer sequences. Statistical analysis was performed using the unpaired, two-tailed $t$-test (Student's $t$-test) or the Mann–Whitney $U$-test.

**Ligand coating and HUVEC-NSC co-culture assays.** Six-well plates were coated with recombinant IgG (11 µg ml⁻¹), Dll4 (1 µg ml⁻¹) or Dll4 plus EGFL7 (10 µg ml⁻¹) at 37 °C. Dll4 was purchased from R&D Systems (Minneapolis, MN, USA); IgG and biological active EGFL7 were purified using a baculovirus system as described previously[22]. After 2 h, plates were washed once with NSC medium and 50,000 3-d-old SVZ-derived neurospheres were seeded in 1 ml NSC medium per well. After 1 h of incubation, cells were collected, lysed, RNA was isolated, quality checked, reverse transcribed into cDNA and then gene expression analysis was performed by qRT-PCR as described above.

Primary HUVECs (P3–P5) were seeded in complete EBM medium at 300,000 cells per 60 mm dish. After 48 h, cells reached 85% confluency and were transfected with siRNA using the Gene Trans II reagent (MoBiTec, Göttingen, Germany) according to the manufacturer's guidelines. The following siRNAs were purchased from Dharmacon (Lafayette, CO, USA): Scrambled control siRNA (**siScr**; GGG AAG CCA UGA ACA ACU U), human EGFL7-specific siRNA (**siEGFL7**; GUG GAU GAA UGC AGU GCU A) and human Dll4-specific siRNA (**siDll4**; CAG AAU GGC UAC UGC AGC A). In brief, 24 µl Gene Trans II was diluted in 156 µl Optimem (Invitrogen) and 9 µl siRNA solution (20 µM) were diluted in 132 µl Diluent B. Both solutions were incubated at RT for 5 min, combined and incubated for another 5 min at RT. HUVECs were washed twice, covered with 2.7 ml Optimem and the Gene Trans II/siRNA solution was added drop-wise. Cells were incubated at 37 °C and 5% $CO_2$ for 4 h. Subsequently, the medium was removed, cells were washed once with PBS and 3 ml fresh, complete EBM medium was added. Upon another 48 h, HUVEC medium was removed and cells were washed once with PBS. A quantity of 70,000 3-d-old SVZ-derived neurospheres were added in complete NSC medium on top of the confluent HUVEC culture. After 1 h of incubation, cells were collected, lysed, RNA was isolated, quality checked, reverse transcribed into cDNA and then gene expression analysis was performed by qRT-PCR as described above. Notch signalling in mouse NSCs was quantified normalizing Hes5 expression to RNA Pol II and GAPDH as murine HKG. The effective knockdown of EGFL7 and Dll4 in the human HUVEC was detected by qRT-PCR using specific probes for EGFL7 and Dll4 and HPRT1 and ACTB as HKG. Each primer pair tested negative for interspecies cross-reactivity.

**In situ hybridization.** Mouse embryos or brains were removed and rapidly rinsed in diethylpyrocarbonate -treated PBS and then fixed in 4% paraformaldehyde-PBS at RT overnight. Tissues were automatically dehydrated in ethanol using a vacuum infiltration processor (Tissue-Tek VIP 5 Jr., Sakura, Alphen aan den Rijn, Netherlands) and embedded in paraffin wax. Sections of 5 μm were prepared using a sliding microtome (SM2000R, Leica, Wetzlar, Germany), mounted on microscope slides and dried at 55 °C overnight. The staining procedure including further processing of the sections such as dewaxing and rehydration was automatically performed by using the Discovery XT System (Roche, Basel, Switzerland). Hybridization of DIG-labelled riboprobes was carried out at 65 °C for 6 h using 50 ng of DIG-labelled riboprobe per slide diluted in 100 μl hybridization buffer. Detection of DIG was performed using an alkaline phosphatase-coupled anti-digoxigenin antibody (Roche) followed by colour development with the use of NBT and BCIP (Roche). Subsequently, cell nuclei were stained with hematoxyline. Slides were mounted with water-soluble mounting medium (Aqua-Poly Mount, Polysciences, Eppelheim, Germany). In situ hybridization slides were analysed and photographed using a BX50 microscope (Olympus, Tokyo, Japan) equipped with a DP72 camera (Olympus).

**Fluorescence in situ hybridization.** Mice were killed by transcardial perfusion with 4% paraformaldehyde-PBS. Brains were removed and post-fixed at 4 °C overnight. Brains were dehydrated in 30% sucrose at 4 °C for 24 h and embedded in Tissue-Tek O.C.T. compound (Sakura), frozen on dry-ice and stored at −80 °C. Cryosections of 10 μm were prepared (Cryostat CM 1900; Leica, Wetzlar, Germany), dried for 30 min at 37 °C and stored at −20 °C until further processing. FISH was performed using QuantiGene ViewRNA ISH Cell Assay (Affymetrix, Santa Clara, CA, USA). For multiplexing Type-1 and Type-6 probes were combined. Images were captured using a confocal microscope (SP8, Leica, Wetzlar, Germany) and quantitative analyses were performed using Imaris 8 (Bitplane, Zurich, Switzerland). To determine the correlation between the levels of Hes5 transcripts and the levels of EGFL7 transcripts, the SVZ was segmented in defined areas of 400 μm$^2$ and the number of transcripts per area was counted (represented as one data point). A Pearson correlation analysis was performed using GraphPad Prism 6.0.

**Immunohistochemistry.** Adult mice were killed by cerebral dislocation. Brains were removed, embedded in Tissue-Tek O.C.T. compound (Sakura), frozen on dry-ice and stored at −80 °C. Cryosections of 10 μm were prepared (Cryostat HM550; Thermo Fisher Scientific, Walldorf, Germany), dried for 1 h at 37 °C and stored at −20 °C until further processing. Subsequent to a PBS wash, sections were incubated with blocking solution (10% goat serum plus 0.01% Triton X-100) at RT for 1 h. Slices were preincubated in mouse-on-mouse serum (M.O.M Kit, Vector Laboratories, Peterborough, UK) in case of mouse-derived antibodies to avoid labelling of endogenous mouse IgG. Sections were incubated with primary antibodies (Supplementary Table 2) in blocking solution at 4 °C overnight. Subsequent to thorough washing, specific staining of antibody was revealed using species-specific fluorophore-conjugated secondary antibodies at RT for 1 h under light protection. Sections were washed in PBS and counterstained using the fluorescent dye 4′,6-diamidino-2-phenylindole (DAPI), applied at a concentration of 10 μg ml$^{-1}$ at RT for 10 min. Sections were washed with distilled water and mounted in Aqua-Poly Mount. Images were captured using a confocal microscope (SP8, Leica, Wetzlar, Germany). The number of marker-positive cells was determined within the first three cell layers adjacent to the ependyma and blotted against the length of the ventricle. All analyses of virus-injected brains were conducted in the hemisphere contralateral to the injection point. Statistical analysis was performed using the unpaired, two-tailed t-test (Student's t-test) or Mann–Whitney U-test between EGFL7 and mock or WT groups.

**Flow-cytometry analyses.** Cell populations were identified by FACS as previously described[2,39,40,64]. Animals were killed by cervical dislocation, brains were taken out and the lateral SVZ was microdissected as a whole mount. To isolate striatal neurons, tissue pieces similar to the SVZ whole-mount were microdissected from the striatum upon SVZ removal. Pooled tissue (5 mice per sample) was digested with Papain and DNAse as previously described[2]. Cells were stained for 20 min on ice in PBS using following antibodies: eFluor405-conjugated rat anti-mCD24 (1:100, BD Pharmingen, San Jose, CA, USA), rabbit anti-mGFAP (1:100, clone G-A-5, Sigma), APC-conjugated rat anti-mGLAST (1:40, clone ACSA-1, Miltenyi, Bergisch Gladbach, Germany), Rhodamine-complexed EGF (1:200, Molecular Probes, Waltham, MA, USA) and eFluor710-conjugated rat anti-mCD133 (1:100, clone 13A4, eBioscience, San Diego, CA, USA). All cell populations were isolated in a single sort using a FACS Aria II (Becton Dickinson, Franklin Lakes, NJ, USA). Data were analysed with FlowJo v9.6.2 software (Tree Star, Ashland, OR, USA). For expression analysis cells were collected in RLT Buffer and RNA extraction was performed as described above.

**SILAC-based mass spectrometry.** Neurosphere protein extracts derived from control and EGFL7$^{-/-}$ cultures were lysed in lysis buffer (4% SDS, 10 mM Tris-HCl, pH 7.5). Protein concentration was measured by the Bradford assay and extracts were mixed in a 1:1 ratio with adult brain protein extracts derived of Lys(6)

( = $^{13}C_6$ lysine)-labelled SILAC mice as an internal protein standard[25]. Proteins were separated by SDS Page and in-gel digestion was performed as described[65]. In brief, each gel lane was divided into equal parts and cut into 1 mm$^2$ cubes. Subsequent to several wash steps, proteins were reduced with 10 mM DTT at 56 °C for 30 min and alkylated with IAA for 30 min at RT in the dark. Gel pieces were washed, dehydrated with ethanol and rehydrated using 50 mM ammonium bicarbonate. 40 μl of an 12 ng μl$^{-1}$ trypsin solution was added and samples were incubated overnight at 37 °C. Generated peptides were extracted using an increasing content of acetonitrile and were subsequently concentrated in a vacuum concentrator. Before liquid chromatography tandem mass spectrometry (LC-MS/MS) analysis, samples were desalted using STAGE tips[66]. Next, peptides were eluted from C18 tips with 30 μl of 0.5% acetic acid in 80% acetonitrile, concentrated in a vacuum concentrator and resuspended in 10 μl buffer A (0.5% acetic acid). The LC-MS/MS equipment consisted of an EASY nLC system coupled to the ion-trap based Orbitrap instrument (Thermo Scientific, Waltham, MA, USA) by a nanospray electron ionization source (Orbitrap XL or Orbitrap Velos). Peptides were separated on an in-house packed 30 cm column (3 μm C18 beads, Dr A. Maisch GmbH, Ammerbuch-Entringen, Germany) using a binary buffer system: A) 0.5% acetic acid and B) 0.1% acetic acid in acetonitrile. Peptides were eluted in a gradient from 5% B to 37% B within 120 min followed by a washing and re-equilibration step. All settings for mass spectrometric analysis are described elsewhere[67]. For raw data processing MaxQuant[68] and the implemented Andromeda search engine were used to correlate the recorded MS/MS spectra against the mouse IPI database. FDR was estimated on the peptide-spectra-match and protein level to 1% by a decoy database approach. Other settings were used as previously described[69].

**Immunoprecipitation and immunoblotting.** Transient transfection of HEK293 cells (ATCC, Middlesex, UK; cell line routinely mycoplasma tested) was performed with Lipofectamine 2000 (Invitrogen) according to the manufacturer's instructions. A total of 3 μg of plasmid DNA was used to transfect cells. Immunoprecipitation and immunoblotting were performed as previously described[70]. In brief, HEK293 cells were lysed in ice-cold lysis buffer containing protease and phosphatase inhibitors. Following centrifugation at 16,200g for 10 min, supernatants were subjected to immunoprecipitation in the presence of 40 μl of Immunosorb A (Medicago, Uppsala, Sweden) overnight at 4 °C on a rotating platform. The following day, the beads were washed 3× in lysis buffer and after centrifugation the bead pellet was resuspended in 20 μl 4× Laemmli protein loading buffer and boiled for 5 min at 95 °C. Immune complexes were loaded onto 10% SDS-polyacrylamide gels and transferred to nitrocellulose membranes (GE Watter and Process, Herentals, Belgium). Membranes were blocked in 10% blocking solution (Rotiblock, Roth, Karlsruhe, Germany) overnight at 4 °C. All primary antibody incubations were performed in blocking buffer for 2 h at RT followed by incubation with fluorescent-labelled anti-mouse antibody for 1 h at RT. The following antibodies were used: mouse anti-V5 (1:2,500, Biozol, Eching, Germany), mouse anti-Flag M2 (1:5,000, Sigma, St. Louis, MI, USA), mouse anti-Tubulin alpha Ab-2 (1:6,000, Thermo Scientific), fluorescent-labelled goat-anti mouse (1:15,000, LI-COR Biosciences, Lincoln, NE, USA). Immunocomplexes were visualized by fluorescence using Odyssey (LI-COR Biosciences) according to the manufacturer's instruction. Image Studio Lite software v4.0 (LI-COR Biosciences) was used for detection and densitometric analysis of western blot data according to manufacturer's instructions. Western blot images in Supplementary Fig. 3 have been cropped for presentation. Full size images are presented in Supplementary Fig. 7.

**Statistical analysis.** Statistical analyses were performed using GraphPad Prism 6.0 software (Statcon, Witzenhausen, Germany). One-way ANOVA, Student's t-test or the Mann–Whitney U-test were used for statistical analysis of usually three biological replicates per experiment; group size was typically between 5 and 10; $P < 0.05$ was considered significant. All experiments were conducted double blinded. F-tests were used to compare variances between groups. Parametric data were analysed using an unpaired two-tailed Student's t-test, in the case of non-parametric data the Mann–Whitney U-test was used instead. Alternatively, one-factor ANOVA was used followed by post hoc analysis for comparisons between three or more groups. Statistical methods used in the analysis are described in figure legends or experimental procedures. No randomization was applied.

**Data availability.** All relevant data are available from the authors.

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

## Acknowledgements

We thank N. Schmarowski and V. Opitz for excellent technical assistance, J. Jordan for MRI data analysis as well as D. O'Neill and C. Ernest for proofreading the manuscript. Furthermore, we thank G. Schütz for providing the Nestin-CreERT2 mouse line. This work was supported by the German Research Foundation (DFG) by the collaborative research center 1080, project A3 (M.H.H.S.), A5 (B.B.) & A6 (B.L.) and the DFG grant SCHM 2159/4-1 to M.H.H.S.

## Author contributions

F.B. performed experiments, analysed data and wrote the manuscript; A.K., F.O., G.H., H.N., J.B., J.H., L.A., N.D.S., P.N.H., R.B., S.K. and V.V. performed experiments and analysed the data; B.B., B.L., K.R., M.K. and T.B. analysed data and contributed to writing of the manuscript; M.H.H.S. designed the study and wrote the manuscript.

## Additional information

**Competing interests:** The authors declare no competing financial interests.

