## [Peer Review File · Nature Communications]

Reviewers' comments:

Reviewer #1 (Remarks to the Author):

This paper examines the role of EGFL7 in modulating SVZ neurogenesis. Using virally-mediated EGFL7 gain of function approaches in vivo and constitutive EGFL7^{-/-} knock-out mouse models, the authors show that EGFL7 exerts a dual role in the SVZ: it suppresses activation of quiescent NSCs and regulates activated NSCs self-renewal and differentiation. They also show that less olfactory neurons are generated in the absence of EGFL7 leading to olfactory behavioural deficits, confirming a role for EGFL7 in neuronal maturation. In vitro experiments and biochemical studies are also presented, which confirm the authors' previous observations of EGFL7 effects in neurosphere cultures and of the Notch binding properties of EGFL7 in an HEK293 overexpression system. Finally, the authors claim that EGFL7 effects on NSC quiescence are dependent on vascular Dll4, largely based on the observation that vascular deletion of Dll4 phenocopies the NSC activation seen in EGFL7^{-/-} mice.

The in vivo characterisation of the EGFL7 gain and loss of function in the SVZ is thorough, and convincing and in itself constitutes an interesting body of work. However, the rest of the manuscript contains several weaknesses that in my opinion greatly limit the novelty and impact of this study. Of particular concern are the extent of data presented, which reproduces previously published results by the authors (Schmidt et al, NCB 2009, PMID:19503073) and the almost complete lack of a mechanistic link between EGFL7 and Dll4. The latter is especially problematic because of the strong claim made by the authors throughout the manuscript of a modulation of Dll4 signalling by EGFL7 at the neurovascular interface. I will discuss these concerns in greater detail below.

Major points:

1. A substantial amount of the in vitro data presented in this manuscript appears to be a repetition of previously published results by the same authors (PMID:19503073). Oddly, the previous work is not even cited in the text when these repeat experiments are presented and many of these experiments are included in main figures as if they were novel findings.

In particular, the following figures/subpanels contain results very similar or sometimes identical to data already published in PMID:19503073:

- a) Fig1e, in situ hybridization of EGFL7 expression in the cortex mirrors the IHC analysis of EGFL7 expression in the cortex previously reported in Fig4c of PMID:19503073 (and with somewhat dissimilar conclusions regarding vascular expression).
- b) Analysis in Fig2b and d is the same as the one presented in Fig3a-c of PMID:19503073
- c) Experiments with EGFL7^{-/-} NSC shown in Fig2e and g are the same as experiments previously presented in PMID:19503073 using siRNA approaches. Although, these should indeed be carried out to confirm that the genetic deletion results in a similar phenotype as the knockdown approach, this data should not be presented in a main figure as it is not a new observation.
- d) Fig3a is a repeat of Fig3d in PMID:19503073
- e) IP experiments shown in Fig3I and S3G were presented in Fig1d-g of PMID:19503073
- f) Fig3M confirms previous results presented in Fig.2a and S7b of PMID:19503073 that EGFL7 competes with jagged1 for Notch binding.

2. Some of the main conclusions of the study are not adequately supported by the data:

- a) No evidence is presented to support the hypothesis that EGFL7 enhances Dll4/Notch signalling in NSC or in the SVZ in vivo at the vascular interface. This conclusion, which is stated in the abstract of

the manuscript, is based entirely on a putative increase in the binding of Notch1 to Dll4 in the presence of EGFL7 in a HEK293 overexpression system and the observation that endothelial deletion of Dll4 in vivo partially phenocopies constitutive EGFL7 deletion. The IP themselves are not very clear due to the lack of labels in Fig.S3i and the somewhat modest differences in Dll4 binding. The results in endothelial-specific Dll4 knock-out mice are interesting, but without additional experiments demonstrating a direct connection between these two molecules, these results are entirely correlative.

If the authors want to claim a role for EGFL7 specifically at the neurovascular interface and through Dll4, this part of the study requires much more work. Evidence should be presented that Dll4-dependent Notch activity is modulated by EGFL7 in NSC and that EGFL7 effects seen in this study are indeed mediated by Dll4. For example, regulation of Notch signalling downstream of Dll4 by EGFL7 should be assessed in NSCs in vitro using recombinant Dll4 ligands and co-culture experiments with endothelial cells. Can the authors show that Dll4 activity is indeed reduced in the absence of EGFL7? Even more important would be a demonstration of a link in vivo. The authors should overexpress EGFL7 in endothelial-specific Dll4 KO animals to determine whether EGFL7 effects are lost in the absence of vascular Dll4 and, conversely, they should determine whether Dll4 overexpression would be sufficient to suppress qNSC activation in EGFL7^{-/-} mice.

b) The model presented in figure 7 is highly speculative and should not be presented. As for the Dll4-mediated effects, there is no evidence in the current manuscript for the potential role for EGFL7 in modulating jagged1 signalling at the neural interface. The idea of a differential effect of EGFL7 at different sites of Notch-mediated cell-cell contacts within the niche is intriguing, but needs to be tested further before such a claim can be made. Given that neurospheres express high levels of jagged1 (FigS3), the authors could perform additional experiments in neurosphere cultures as a proxy for contacts between aNSC and TAPs to test their model. In addition their beautiful lineage tracing method could be used to assess effects on self-renewal and division modes downstream of EGFL7/jagged1 regulation. This would be particularly interesting given the previous findings by Basak et al (J. Neuroscience 2012) of a selective role for Notch1 in the regulation of activated NSC in vivo.

Minor points:

The text does not match the results in line 233 which should read tamoxifen increased the amount of proliferating cells

Reviewer #2 (Remarks to the Author):

The components of the adult subventricular zone neural stem cell niche and the process of regulating stem cell activation and neurogenesis are not well understood. There are a number of reports showing the participation of the vasculature and blood-borne factors in the control of neurogenesis in the adult forebrain. This topic is by no means exhausted, or even clear. In this manuscript the authors address the role of a putative non-canonical Notch ligand EGFL7 in the regulation of neural stem cell activity in the adult subventricular zone. The authors show that EGFL7 controls the activity of adult neural stem cells and their data support that gain and loss of EGFL7 affects proliferation and neurogenesis. The authors use an impressive array of in vivo and in vitro experiments to substantiate this. They claim that the mode of EGFL7 action is through the regulation of Notch signaling, a function that this group has already shown. They use biochemical assays to show that EGFL7 changes the binding affinity of Notchs for Dll4 and Jagged1 and go on to claim that this affects NSCs activity. This is an interesting manuscript using an impressive array of techniques and the data are, in the main, solid.

Major comments.

1. Some of the in vivo findings are not supported by the in vitro experiments and seem to be even contradictory, at least in terms of Notch activity. The authors should provide solid explanations for these discrepancies.
2. Why and how should Jagged1 and Dll4 result in different effects which, from biochemical analysis and binding affinity assays, the authors imply are through Notch1? The authors need to address or discuss this in detail.
3. The EGFL7-null mouse phenotype in the adult SVZ could have a development aspect that the authors should clarify by analyzing neurogenesis in the early forebrain during the perinatal period or at least discuss in detail.
4. The authors should perform a more detailed analysis of cell cycle using BrdU pulse chase and/or double Thymidine analogue incorporation assays to assess changes in cell cycle length.
5. Much of the data (increased cells in cell cycle and reduced neuron production) could be interpreted as a block or delay in NSC/progenitor differentiation, which the authors mention in the discussion but which should be elaborated upon more.
6. Figure 6: Why does EGFL7 have such different effects here? GOF leads to increase in neurons and increased proliferation, LOF reduced neurons. This needs to be clarified either with discussion or data.

Minor comments.

7. The quality of the fluorescent images in Figures 1G-H and 3E-J needs to be improved. The inserts are too small and the signal barely detectable.
8. Line 432: What is meant by "...keep differentiated NSCs quiescent...". This does not make sense to me and should be changed.
9. Careful editing by a native English speaker would help the readability of the manuscript.

Reviewer #3 (Remarks to the Author):

Bicker and colleagues investigate the role of the non-canonical Notch ligand EGFL7 on SVZ stem cells and their progeny. Using a variety of in vivo and in vitro approaches, they investigate the effect of gain of function and loss of function of EGFL7 on adult neural stem cells and their daughter cells, as well as how this impacts olfactory bulb functional output and behavior. While the authors use many different approaches, the analysis with any one approach is sometimes quite perfunctory. There are several major points that warrant additional experiments to support some of the conclusions of this study. Moreover, it would also be helpful for the reader if only relevant data are presented, as the manuscript is sometimes difficult to follow. Many low power images are difficult to interpret, and could be removed. Higher power images, and orthogonal views would be more convincing. Figure legends often do not match or describe the experiments shown.

1- An important point of the paper is that deletion of EGFL7 modulates aNSC behaviour, and overexpression of EGFL7 pushes aNSC to a quiescent state. Given the potential role of EGFL7 modulating quiescence and activation of adult NSC, it is surprising that although the authors first characterize the expression of EGFL7 in purified populations, including quiescent NSC, they do not characterize the quiescent stem cell pool in most of their experiments. All in vivo analysis is based on GFAP+Nestin+ or GFAP+Nestin+Ki67+ immunostaining, but analysis of GFAP+Nestin-S100b- (quiescent neural stem cells) or differentiated astrocytes (GFAP+Nestin-S100b+) is not performed. qNSC are only analyzed by FACS in Figure 5. It is important to quantify the effects on qNSCs, as well as differentiated astrocytes in the various experiments. Indeed in a previous paper, they showed that modulation of EGFL7 (Mirko et al, 2009) affected neuronal and oligodendrocyte formation at the expense of the astrocytes. Is there a change in oligodendrocyte formation in vivo?

2- To examine the effects of EGFL7 on stem cell properties in vitro, the authors perform in vitro assays with bulk V-SVZ cells. The authors' conclusions regarding stemness and activation from these in vitro assays are difficult to interpret using bulk SVZ cells. Multiple populations can give rise to neurospheres in vitro (primarily aNSC/TAC). Given the ability of authors to FACS purify SVZ cells, the study would be strengthened if the response of purified populations was examined. Importantly, the measure of size is not an accurate measure of "stemness", as even at low densities, cells or spheres readily aggregate and fuse. Size is a reflection of how much cells in the spheres divide. The authors should distinguish between effects on activation (number of clones that form) and growth/size (proliferation of individual clones). Single cell assays of FACS purified cells would allow them to monitor and distinguish the effect on two parameters at the same time.

3-The EGFL7-KO model is constitutive. As EGFL7 is expressed from early on during development, not all the changes described might be a result of changes in adult SVZ. Are there changes at earlier stages of development? Is blood vessel architecture maintained in the mutants? If not, are there any issues with altered permeability?

4- Throughout the paper, the figure legends and text and claims do not always match. One example is the statement in the abstract that metabolism of cells is affected, yet in the Silac experiments, only synaptic data are presented in the figure, nothing about metabolism, although the figure legend mentions it.

5- Statistics. The authors use t-test as statistical methods except ANOVA in behavioural studies. In most cases, the data is presented as fold change normalized to WT. Non-parametric tests would be more accurate (Mann-Whitney test).

Figure 7 should be revised in light of comments.

Please find below our responses to the the reviewer's comments.

Reviewer #1: Remark to the author...

1. A substantial amount of the *in vitro* data presented in this manuscript appears to be a repetition of previously published results by the same authors (PMID:19503073). Oddly, the previous work is not even cited in the text when these repeat experiments are presented and many of these experiments are included in main figures as if they were novel findings. In particular, the following figures/subpanels contain results very similar or sometimes identical to data already published in PMID:19503073:

Response to the reviewer

We thank the reviewer for his careful revision. In order to explain the partial redundancy observed we'd like to quote the cover letter to the editor that accompanied the initial submission of our manuscript:

"Our work is based on our study published in Nature Cell Biology in 2009, in which we have shown that EGFL7 affects Jagged-induced Notch signaling in NSCs *in vitro*. In the following 7 years we worked on the validation and expansion of our findings *in vivo*, and the current manuscript is a follow-up study on this particular paper."

There was no intent to hide that our manuscript follows up on our 2009 study in Nature Cell Biology that identified EGFL7 as a novel non-canonical Notch ligand and neural stem cell regulator. On the contrary, we emphasized it in the submission cover letter. Our past work mostly relied on *in vitro* techniques; therefore, it seemed essential to substantiate our findings *in vivo*. However, as a follow-up study our present manuscript is intimately linked to our past work and some data points do indeed overlap. We tried to minimize these interlinks but could not fully delete them as it would require the reader to read both articles in parallel to get the full picture. We are sorry if this approach did not adequately highlight that this manuscript is closely linked with our 2009 NCB publication. Furthermore, some open questions raised in 2009 have been picked up in the current

work and have been answered using novel state-of-the-art tools and techniques. Lastly, the manuscript is structured in a way that the EGFL7 gain- and loss-of-function models are opposing each other in various figures. Occasionally, this required the repetition of a previous experiment with the sole intent to present the complete picture in this manuscript.

In order to address the reviewer's concerns, various data points have been moved to the supplement and we now refer more frequently to our 2009 NCB in the appropriate text passages. Further details are discussed below.

a) Fig1e, in situ hybridization of EGFL7 expression in the cortex mirrors the IHC analysis of EGFL7 expression in the cortex previously reported in Fig4c of PMID:19503073 (and with somewhat dissimilar conclusions regarding vascular expression).

Response to the reviewer

The reviewer's point is well taken. Our previous IHC did not allow the cellular source of EGFL7 to be identified as it is a secreted protein. Therefore, *in situ* hybridizations have been performed to detect the mRNA of EGFL7. Additionally, this technique was much more sensitive than the previous IHCs. Results obtained confirmed neurons as a primary cellular source of EGFL7 but thanks to our improved technique also a vascular EGFL7 signal could now be detected.

b) Analysis in Fig2b and d is the same as the one presented in Fig3a-c of PMID:19503073

Response to the reviewer

Indeed, these were confirmations of previous experiments presented in order to complete the corresponding data obtained from EGFL7^{-/-} mice. In order to address the reviewer's concern, both plots have been moved to the supplement (Suppl. Figs. 2D+E) and the phrase "...as previously described (Schmidt et al., 2009)." has been added to the text.

c) Experiments with EGFL7^{-/-} NSC shown in Fig2e and g are the same as experiments previously presented in PMID:19503073 using siRNA approaches. Although, these should indeed be carried out to confirm that the genetic deletion results in a similar phenotype as the knockdown approach, this data should not be presented in a main figure as it is not a new observation.

Response to the reviewer

In order to address the reviewer's concern, both plots have been moved to the supplement (Suppl. Figs. 2F+G) and the sentence "..., confirming previous data obtained by knock-down approaches (Schmidt et al., 2009)" has been added to the manuscript.

d) Fig3a is a repeat of Fig3d in PMID:19503073

Response to the reviewer

Indeed, this particular piece of information was published in our 2009 NCB and was appropriately cited in the text. However, it is just a small piece of data in a larger data set and needed at this place to illustrate the opposing effects of EGFL7 *in vitro* and *in vivo*. Currently, it can easily be appreciated from Figure 3 that *Hes5* (as a marker for Notch signaling) drops down *in vitro* and goes up *in vivo* once EGFL7 is present. Moving the data presented in Figure 3A to the supplement would mask the information, therefore we prefer to leave the subfigure in place. However, in order to address the reviewer's concern the phrase "...as previously presented." has been added to the text describing this subfigure.

e) IP experiments shown in Fig3I and S3G were presented in Fig1d-g of PMID:19503073

Response to the reviewer

The point of the reviewer is well taken, however, Fig 3I is not just a repetition of previous experiments. This novel data set is a quantitative assessment of the binding strength of EGFL7 to the extracellular domains of the four Notch receptors. Previously, only the binding of EGFL7 to the four Notch receptors *per se* had been demonstrated. Now it can be judged, which of the Notch receptors preferentially binds EGFL7, which allows for the conclusion of a binding hierarchy. This is especially important as it is more and more recognized in the field that the four Notch receptors do not just mediate redundant functions. In order to obtain this data the extracellular domains of Notch1-4 (previously, the full length receptors were analyzed) were cloned in the same expression vector with identical tags (previously, different tags were used). Further, an Odyssey system has been used for the quantitative analysis of the western blot data, which was not possible in our previous work. Together, this makes the experiment unprecedented. Nevertheless, the Figure has been moved to the supplement to address the reviewer's concern (Suppl. Fig. 3M).

f) Fig3M confirms previous results presented in Fig.2a and S7b of PMID:19503073 that EGFL7 competes with jagged1 for Notch binding.

Response to the reviewer

Indeed, our previous finding on the competition of EGFL7 and Jagged1 for Notch1 binding was confirmed but additionally quantified using novel expression constructs. However, the Jagged1-Notch1 interaction is presented only as a control in this particular experimental setting. The central message of the subfigure is that EGFL7 cooperates with Dll4 but competes with Jagged1 in terms of Notch1 binding. This antagonistic attitude of

EGFL7 towards the two canonical Notch ligands is key to understanding the impact of EGFL7 on Notch signaling. In order to show this antagonism, the impact of EGFL7 on both the Dll4-Notch1 as well as the Jagged1-Notch1 interaction needs to be presented.

Reviewer #1: Remark to the author...

2. *Some of the main conclusions of the study are not adequately supported by the data:*

a) No evidence is presented to support the hypothesis that EGFL7 enhances Dll4/Notch signalling in NSC or in the SVZ in vivo at the vascular interface. This conclusion, which is stated in the abstract of the manuscript, is based entirely on a putative increase in the binding of Notch1 to Dll4 in the presence of EGFL7 in a HEK293 overexpression system and the observation that endothelial deletion of Dll4 in vivo partially phenocopies constitutive EGFL7 deletion. The IP themselves are not very clear due to the lack of labels in Fig.S3i and the somewhat modest differences in Dll4 binding. The results in endothelial-specific Dll4 knock-out mice are interesting, but without additional experiments demonstrating a direct connection between these two molecules, these results are entirely correlative.

If the authors want to claim a role for EGFL7 specifically at the neurovascular interface and through Dll4, this part of the study requires much more work. Evidence should be presented that Dll4-dependent Notch activity is modulated by EGFL7 in NSC and that EGFL7 effects seen in this study are indeed mediated by Dll4. For example, regulation of Notch signalling downstream of Dll4 by EGFL7 should be assessed in NSCs in vitro using recombinant Dll4 ligands and co-culture experiments with endothelial cells. Can the authors show that Dll4 activity is indeed reduced in the absence of EGFL7? Even more important would be a demonstration of a link in vivo. The authors should overexpress EGFL7 in endothelial-specific Dll4 KO animals to determine whether EGFL7 effects are lost in the absence of vascular Dll4 and, conversely, they should determine whether Dll4 overexpression would be sufficient to suppress qNSC activation in EGFL7^{-/-} mice.

Response to the reviewer

The reviewer made a number of excellent suggestions of additional experiments to improve our manuscript, which we followed in order to support our hypothesis. The following additional assays have been performed and added to the results and discussion sections of the manuscript:

- 1) Plastic dishes have been coated with recombinant purified protein using IgG as a negative control and Dll4 in the absence or presence of EGFL7. Subsequently, neurospheres have been seeded onto these dishes and a *Hes5* qRT-PCR has been performed to quantify the effect on Notch signaling in NSCs. Indeed, Dll4-induced Notch signaling was increased in the presence of EGFL7 (Suppl. Fig. 3J).
- 2) Primary human umbilical vein endothelial cells (HUVECs) have been treated with control siRNA or siRNAs specific for Dll4 and EGFL7 alone and in combination. Subsequently, neurospheres have been seeded on top and a mouse-specific *Hes5* qRT-PCR has been performed to quantify Notch signaling in the co-cultured NSCs, which was reduced upon EGFL7 and Dll4 knock-down in endothelial cells in an additive manner (Suppl. Fig. 3K).
- 3) The amount of neurosphere initiating cell-derived spheres (NICs) *ex vivo* was determined as a measure of the amount of aNSCs *in vivo* and was found to be increased upon tamoxifen-induced Dll4 knock-out in *Dll4^{fl/fl};Cdh5(PAC)-CreERT2* mice (Fig. 3M).
- 4) In order to demonstrate a link between EGFL7 and Dll4 in NSC regulation *in vivo*, CVI of AdEGFL7 into the ventricle of *Dll4^{fl/fl};Cdh5(PAC)-CreERT2* mice has been performed and the number of primary neurospheres forming from infected SVZ tissue has been counted. AdEGFL7 reduced the amount of NICs as expected, but this effect was reduced in the absence of vascular Dll4 (Fig. 3M). Conversely, CVI of AdDll4 into the ventricle of EGFL7 WT mice reduced the amount of NICs but the effect was weaker in EGFL7^{-/-} animals (Fig. 3N). In conclusion, these findings substantiated our hypothesis of EGFL7 and Dll4 cooperating in NSC regulation.
- 5) Dll4 has been knocked-out in stem cells using the novel *Dll4^{fl/fl};Nestin-CreERT2* mouse model. Loss of neural Dll4 did not affect primary neurospheres, supporting our

hypothesis that it is the vascular Dll4 which is most relevant for NSC regulation (Suppl. Fig. 3I).

Reviewer #1: Remark to the author...

b) The model presented in figure 7 is highly speculative and should not be presented. As for the Dll4-mediated effects, there is no evidence in the current manuscript for the potential role for EGFL7 in modulating jagged1 signalling at the neural interface. The idea of a differential effect of EGFL7 at different sites of Notch-mediated cell-cell contacts within the niche is intriguing, but needs to be tested further before such a claim can be made. Given that neurospheres express high levels of jagged1 (FigS3), the authors could perform additional experiments in neurosphere cultures as a proxy for contacts between aNSC and TAPs to test their model. In addition their beautiful lineage tracing method could be used to assess effects on self-renewal and division modes downstream of EGFL7/jagged1 regulation. This would be particularly interesting given the previous findings by Basak et al (J. Neuroscience 2012) of a selective role for Notch1 in the regulation of activated NSC in vivo.

Response to the reviewer

We agree with the reviewer that a thorough analysis of a conditional Jagged 1 knock-out mouse model in various tissues would support our model. Although the excellent work performed by Ottone et al. (Nat Cell Biol, 2014) and Nyfelder et al. (Embo J, 2005) on this topic have been incorporated in our cartoon, a full characterization of the Jagged 1 knock-out, as we present it here for EGFL7, is not yet available. However, the creation and characterization of such a model from scratch will take years and is beyond the scope of our manuscript. We intended to make the point that EGFL7 acts at the neurovascular interface by the modulation of Dll4-induced Notch signaling. Jagged 1 was not the center of our attention as it is quite ubiquitously expressed and not restricted to the neurovascular interface. Most likely, a conditional Jagged 1 knock-out mouse model will show a plethora of phenotypes difficult to interpret and is beyond what can be done for this particular revision. In order to address the reviewer's concern the cartoon has been

removed from the manuscript.

Reviewer #1: Remark to the author...

Minor points:

The text does not match the results in line 233 which should read tamoxifen increased the amount of proliferating cells.

Response to the reviewer

This typo has been corrected.

Reviewer #2: Remark to the author...

Major comments.

1. Some of the in vivo findings are not supported by the in vitro experiments and seem to be even contradictory, at least in terms of Notch activity. The authors should provide solid explanations for these discrepancies.

Response to the reviewer

We are grateful for the reviewer's constructive comments. We hypothesized that *Hes5* signaling in neurospheres *in vitro* and neural stem cells *in vivo* was different due to the canonical Notch ligand Dll4, which is expressed on blood vessels *in vivo* but absent from neurospheres *in vitro*. To make this clearer, the discussion (Pg 21, bottom) has been changed to read (in part):

"However, the influence EGFL7 exerted on Notch signaling as measured by *Hes5* levels in neurospheres and SVZ tissue was, in part, in opposition. [...] We hypothesized that EGFL7 may exerted its effect on *Hes5* via Notch signaling components differentially expressed in

SVZ tissue and neurospheres. An expression screen of various Notch signaling components identified the Notch ligand Dll4 as a potential niche component that was expressed on blood vessels but absent from neurospheres.”

2. *Why and how should Jagged1 and Dll4 result in different effects which, from biochemical analysis and binding affinity assays, the authors imply are through Notch1? The authors need to address or discuss this in detail.*

Response to the reviewer

The reviewer is completely right that the interpretation of these interaction data is quite complex as the canonical Notch ligands Jagged1 and Dll4 bind to Notch1 in two different ways. Either *in trans* in between two cells, which activates Notch signaling, or *in cis* on the same cell, which inhibits Notch signaling. Our interaction analyses have shown that EGFL7 bound the EGF-like repeats 11+12 in Notch1, which are responsible for Notch1 activation *in trans* but not the EGF-like repeats 24-29 responsible for Notch1 inhibition *in cis*. Therefore we concluded that EGFL7 may only affect Notch1 activation *in trans*.

In the neurovascular stem cell niche, the interaction between Dll4 and Notch1 may only occur *in trans* as Dll4 expression is restricted to blood vessels, therefore, it can only activate Notch1 on the surface of neighboring neural stem cells but not inhibit it. Jagged1, however, is ubiquitously expressed and can either activate or inhibit Notch signaling in neural stem cells. Moreover, we found that EGFL7 increased the binding of Dll4 to Notch1 but decreased the binding of Jagged1 to Notch1.

A combination of both observations allowed for the conclusion that high levels of EGFL7 *in vivo* inhibited Jagged1 binding to Notch1 *in trans* but this loss was compensated by an increased recruitment of the ligand Dll4. Therefore, increased levels of EGFL7 *in vivo* caused an increased recruitment of Dll4 and as a result increased levels of Notch signaling as experimentally verified. However, in neurospheres *in vitro*, Dll4 was not expressed and no compensation of the Jagged1 loss could occur. Consequently, EGFL7 blocked the

Jagged1-Notch1 interaction *in trans* (activation) but did not affect binding *in cis* (inactivation) whereby only the Notch inhibiting potential of Jagged1 remained. As a consequence, increased amounts of EGFL7 *in vitro* converted the Notch1 ligand Jagged1 into an inhibitor of Notch signaling (Schmidt et al., Nat Cell Biol, 2009). In sum, the differential effect EGFL7 has on Notch signaling *in vitro* and *in vivo* results from a combination of its preferential binding site in the Notch1 receptor and the differential effects it exerts on the Notch ligands Dll4 and Jagged1.

In order to address the reviewer's concern the respective paragraph (Pg. 22) in the discussion of the manuscript has been largely extended to include the explanation provided above.

3. The EGFL7-null mouse phenotype in the adult SVZ could have a development aspect that the authors should clarify by analyzing neurogenesis in the early forebrain during the perinatal period or at least discuss in detail.

Response to the reviewer

The reviewer is correct that a developmental mismatch in EGFL7^{-/-} mice might be responsible for the deficits observed in adult neurogenesis. In order to rule out this possibility, several test assays have been performed as discussed in the manuscript. First, the total amount of stem cells per slide was not different in wild-type and knock-out animals (Suppl. Fig. 4J). Second, MRI studies revealed no gross differences in brain, olfactory bulb or ventricle size in wild-type or knock-out animals (Suppl. Fig. 6A). Furthermore, data derived from the novel tamoxifen-inducible, conditional EGFL7 knock-out mouse model *EGFL7^{fl/fl};Nestin-CreERT2* is now presented as Suppl. Fig. 4K+L and Fig. 6H. Indeed, the tamoxifen-induced knock-out of EGFL7 from adult neural stem cells phenocopied the results obtained in the conventional EGFL7^{-/-} model, indicating that defects observed in adult neurogenesis were not due to a developmental malformation of the murine brain in EGFL7^{-/-} mice.

4. *The authors should perform a more detailed analysis of cell cycle using BrdU pulse chase and/or double Thymidine analogue incorporation assays to assess changes in cell cycle length.*

Response to the reviewer

In order to address the reviewers concern a FACS-based cell cycle analysis of EGFL7^{-/-}-derived neurospheres has been performed. Data revealed an increased amount of NSCs in the G₂/M phase, which was achieved at the expense of cells in the G₀/G₁ phase. Data is now presented on page 10 and in Fig. 2E+F.

5. *Much of the data (increased cells in cell cycle and reduced neuron production) could be interpreted as a block or delay in NSC/progenitor differentiation, which the authors mention in the discussion but which should be elaborated upon more.*

Response to the reviewer

In order to address the reviewer's concern the discussion on this topic (Pg. 21) has been expanded and reads now as follows: " Comparison of the transcriptomes of neurospheres derived from EGFL7^{-/-} and WT mice by SILAC-based mass spectrometry revealed that EGFL7^{-/-} spheres experienced a delay in differentiation *in vitro*. WT neurospheres displayed a type of differentiation at d5 in culture and upregulated proteins responsible for neuronal signaling, e.g., glutamate receptors or synaptic proteins. EGFL7^{-/-} spheres, however, retained a protein expression profile at d5 similar to d2 in culture which is indicative of increased metabolic activity²⁵ with high levels of riboproteins and nucleic acid metabolism enzymes. We hypothesize that delayed differentiation and sustained proliferation of EGFL7^{-/-} NSCs caused larger neurospheres as observed *in vitro*."

6. *Figure 6: Why does EGFL7 have such different effects here? GOF leads to increase in neurons and increased proliferation, LOF reduced neurons. This needs to be clarified either with*

discussion or data.

Response to the reviewer

We'd like to point out the following paragraph within the discussion (Pg. 23):

“Indeed, ectopic expression of EGFL7 pushed aNSCs towards differentiation into TAPs, NBs and eventually interneurons. This was probably due to the transient inhibition of Jagged1-mediated Notch signaling in aNSCs and therefore comparable to the situation in RBPJ^{-/-} mice¹⁶, where the induced knock-out of this central Notch signaling component led to a transient activation of neurogenesis in the adult SVZ. Conversely, loss of EGFL7 caused an accumulation of aNSCs within the SVZ and a smaller proportion of them entered the neuronal differentiation pathway. Consequently, less interneurons were formed in the OB of adult EGFL7^{-/-} mice. This indicates that EGFL7 is necessary for neuronal differentiation of NSCs/NPCs but also affects the activity state of NSCs.”

Furthermore, the paragraph in the discussion analyzing Fig. 6 (Pg. 24) has been modified in order to address the reviewer's concern in the following manner:

“In order to define whether or not the regulation of NSCs/NPCs by EGFL7 is of physiological relevance for the adult brain, the amount of adult-born neurons in the OB was determined and was found to be reduced in EGFL7^{-/-} but increased in AdEGFL7-infected mice, due to the Notch-dependent alteration of neurogenesis in the SVZ (see above).”

Minor comments.

7. *The quality of the fluorescent images in Figures 1G-H and 3E-J needs to be improved. The inserts are too small and the signal barely detectable.*

Response to the reviewer

The size of the Figures has been increased as suggested by the reviewer.

8. Line 432: What is meant by "...keep differentiated NSCs quiescent....". This does not make sense to me and should be changed.

Response to the reviewer

This particular sentence was changed into: "Apparently, EGFL7 was needed to keep NSCs quiescent as EGFL7^{-/-} cells re-entered the cell cycle more frequently."

9. Careful editing by a native English speaker would help the readability of the manuscript.

Response to the reviewer

The revised manuscript has now been seen by two additional native speakers.

Reviewer #3: Remark to the author...

Bicker and colleagues investigate the role of the non-canonical Notch ligand EGFL7 on SVZ stem cells and their progeny. Using a variety of in vivo and in vitro approaches, they investigate the effect of gain of function and loss of function of EGFL7 on adult neural stem cells and their daughter cells, as well as how this impacts olfactory bulb functional output and behavior. While the authors use many different approaches, the analysis with any one approach is sometimes quite perfunctory. There are several major points that warrant additional experiments to support some of the conclusions of this study. Moreover, it would also be helpful for the reader if only relevant data are presented, as the manuscript is sometimes difficult to follow. Many low power images are difficult to interpret, and could be removed. Higher power images, and orthogonal views would be more convincing. Figure legends often do not match or describe the experiments shown.

Response to the reviewer

We are grateful for these constructive comments and have reworked the manuscript according to the reviewer's suggestions. In particular, images have been improved and figure legends have been screened for mismatches. Furthermore, the manuscript has been clarified by the removal of marginal data sets to address the reviewer's suggestion to simplify the manuscript. Specifically, former Figures 2H-I and 6H have been deleted, as data presented were interesting side observations but did not directly relate to the main topic of the manuscript. Additionally, former Fig. 5 L has been moved to Fig. 2 in order to present all data on neural stem cells *in vitro* in one figure.

Reviewer #3: Remark to the author...

1- An important point of the paper is that deletion of EGFL7 modulates aNSC behaviour, and overexpression of EGFL7 pushes aNSC to a quiescent state. Given the potential role of EGFL7 modulating quiescence and activation of adult NSC, it is surprising that although the authors first characterize the expression of EGFL7 in purified populations, including quiescent NSC, they do not characterize the quiescent stem cell pool in most of their experiments. All in vivo analysis is based on GFAP+Nestin+ or GFAP+Nestin+Ki67+ immunostaining, but analysis of GFAP+Nestin-S100b- (quiescent neural stem cells) or differentiated astrocytes (GFAP+Nestin-S100b+) is not performed. qNSC are only analyzed by FACS in Figure 5. It is important to quantify the effects on qNSCs ... in the various experiments.

Response to the reviewer

In order to address the reviewer's concern the populations of quiescent neural stem cells (GFAP⁺Nestin⁻S100b⁻) and differentiated astrocytes (GFAP⁺Nestin⁻S100b⁺) in the SVZ have been quantified by IF upon AdEGFL7 CVI and in EGFL7^{-/-}. Data is now presented as Fig. 5K+L and Suppl. Fig. 5J+K. Indeed, ectopic expression of EGFL7 increased the amount of

qNSCs, while EGFL7 knock-out animals displayed a lower amount of them. The total amount of astrocytes in the SVZ, however, remained unaltered.

Reviewer #3: Remark to the author...

It is important to quantify the effects on ... differentiated astrocytes in the various experiments. Indeed in a previous paper, they showed that modulation of EGFL7 (Mirko et al, 2009) affected neuronal and oligodendrocyte formation at the expense of the astrocytes. Is there a change in oligodendrocyte formation in vivo?

Response to the reviewer

This is an excellent point but neither an increased amount of newborn astrocytes nor oligodendrocytes has been observed in the SVZ, RMS or the OB, tissues that were in the focus of our current study (Suppl. Fig. 5J+K and data not shown). However, adult-born OPCs migrate from the ventricle towards the corpus callosum, a structure that we have not yet analyzed. This is definitely a fascinating topic, which we plan to tackle upon the availability of additional tools, such as a conditional EGFL7 knock-out model in OPCs ($EGFL7^{fl/fl};NG2-CreERT2$), which we currently produce. However, to study the impact of EGFL7 on OPCs and oligodendrocyte maturation might take a year or more as it is a completely new project. Therefore, it is in our opinion beyond the scope of the current manuscript, in which we try to unravel the implications of EGFL7 in adult neurogenesis rather than in gliogenesis.

Reviewer #3: Remark to the author...

2- To examine the effects of EGFL7 on stem cell properties in vitro, the authors perform in vitro assays with bulk V-SVZ cells. The authors' conclusions regarding stemness and activation from these in vitro assays are difficult to interpret using bulk SVZ cells. Multiple populations can give rise to neurospheres in vitro (primarily aNSC/TAC). Given the ability of authors to FACS purify SVZ cells, the study would be strengthened if the response of purified populations was examined.

Importantly, the measure of size is not an accurate measure of "stemness", as even at low densities, cells or spheres readily aggregate and fuse. Size is a reflection of how much cells in the spheres divide. The authors should distinguish between effects on activation (number of clones that form) and growth/size (proliferation of individual clones). Single cell assays of FACS purified cells would allow them to monitor and distinguish the effect on two parameters at the same time.

Response to the reviewer

This is a great suggestion and we verified at the beginning of the project that our assays yielded comparable results no matter if neurospheres were cultured at clonal density or in single wells (added to Results on Pg. 8 and as Suppl. Fig. 2C). However, occasionally, we observed that the sorting process yielded NSC populations with up to a five-fold higher self-renewal capacity as compared to neurospheres cultured at clonal density. This indicated to us that the sorting process itself may affect the NSC population, which was to be analyzed. This is not problematic for straight FACS analysis as performed in the current manuscript but subsequent biological paradigms, such as the neurosphere assay, might be affected by the sorting procedure. Furthermore, each sorting procedure requires the application of specific markers and depending on marker and/or antibody selected, inherits some specific bias, which we tried to avoid whenever possible. For these two reasons we decided to perform the neurosphere assays in this manuscript in a "classical" style at clonal density.

Furthermore, we fully agree that the term "stemness" is misleading. Our intended use was – as it occurs in the literature – to describe size measurements of neurospheres. However, throughout the manuscript stemness has been equated with proliferation and self-renewal with activation as interpreted by the reviewer. To resolve this, the term stemness has now been replaced by the term size throughout the manuscript.

Reviewer #3: Remark to the author...

3-The EGFL7-KO model is constitutive. As EGFL7 is expressed from early on during development, not all the changes described might be a result of changes in adult SVZ. Are there changes at earlier stages of development? Is blood vessel architecture maintained in the mutants? If not, are there any issues with altered permeability?

Response to the reviewer

Indeed, a developmental mismatch in EGFL7^{-/-} mice might be responsible for the deficits observed in adult neurogenesis. However, this would not apply to the EGFL7 gain of function studies performed here. But in order to rule out the possibility of developmental defects a couple of additional test assays have been performed. First, the total amount of stem cells per slide was not different in wild-type and knock-out animals (Suppl. Fig. 4J). Second, MRI studies revealed no gross differences in brain, olfactory bulb or ventricle size in wild-type and knock-out animals (Suppl. Fig. 6A). Furthermore, the reviewer is completely right that the blood vessels in the SVZ might be altered in the conventional EGFL7^{-/-} mice. Immunohistochemical analyses using CD31 as a blood vessel marker did not show any gross abnormalities of the blood vessels in the SVZ (data not shown). However, in order to fully address the reviewer's concern a novel tamoxifen-inducible, conditional EGFL7 knock-out mouse model has been established and results are now presented as Suppl. Fig. 4K+L and Fig. 6H. The specific EGFL7 knock-out in adult neural stem cells, using an EGFL7^{fl/fl};Nestin-CreERT2 model, phenocopied the results obtained in the constitutive EGFL7^{-/-} model. Data indicate that the defects observed in adult neurogenesis in EGFL7^{-/-} mice are stem cell specific but not related to developmental or blood vessel malformations.

Reviewer #3: Remark to the author...

4- Throughout the paper, the figure legends and text and claims do not always match. One example is the statement in the abstract that metabolism of cells is affected, yet in the Silac

experiments, only synaptic data are presented in the figure, nothing about metabolism, although the figure legend mentions it.

Response to the reviewer

Though the point of the reviewer is well-taken, this impression might be due to a misleading phrasing of the Figure legend. We intended to make the point that wild-type neurospheres upregulate neuronal signaling proteins at d5 in culture. These are the ones depicted in Fig. 2D. EGFL7^{-/-} spheres, however, retained a protein expression profile at d5 which was comparable to d2 with high levels of riboproteins and metabolic enzymes. We concluded that EGFL7^{-/-} spheres exhibited delayed differentiation but sustained proliferation, causing larger spheres.

In order to address the reviewer's concern this part of the figure legend has been rephrased and now reads as follows: "SILAC-based mass spectrometric proteome analyses of EGFL7^{-/-} and WT neurospheres was performed after 2 and 5 d in culture. WT spheres upregulated proteins involved in neuronal signaling, e.g., glutamate receptors or synaptic proteins at d5, suggesting differentiation *in vitro*. However, EGFL7^{-/-} spheres displayed limited upregulation of these proteins but instead, retained an expression profile at d5 that was comparable to WT and KO spheres at d2 in culture with high levels of riboproteins and metabolic enzymes. This indicates that EGFL7^{-/-} spheres displayed delayed differentiation but sustained proliferation *in vitro* resulting in larger spheres."

Furthermore, the paragraph in the discussion (Pg. 21) dealing with this topic has been expanded: "Comparison of the transcriptomes of neurospheres derived from EGFL7^{-/-} and WT mice by SILAC-based mass spectrometry revealed that EGFL7^{-/-} spheres experienced a delay in differentiation *in vitro*. WT neurospheres displayed a type of differentiation at d5

in culture and upregulated proteins responsible for neuronal signaling, e.g., glutamate receptors or synaptic proteins. EGFL7^{-/-} spheres, however, retained a protein expression profile at d5 similar to d2 in culture which is indicative of increased metabolic activity²⁵ with high levels of riboproteins and nucleic acid metabolism enzymes. We hypothesize that delayed differentiation and sustained proliferation of EGFL7^{-/-} NSCs caused larger neurospheres as observed *in vitro*.”

Reviewer #3: Remark to the author...

5- Statistics. The authors use t-test as statistical methods except ANOVA in behavioural studies. In most cases, the data is presented as fold change normalized to WT. Non-parametric tests would be more accurate (Mann-Whitney test).

Response to the reviewer

The reviewer’s point is well taken and we are grateful for this suggestion. In order to address the reviewer’s concern all calculations have been repeated using the Mann Whitney *U* test, however, results remained significant as indicated in the text.

Reviewer #3: Remark to the author...

Figure 7 should be revised in light of comments.

Response to the reviewer

Due to concerns of both reviewer 3 and 1, Figure 7 has been removed from the
Page 19 of 20

manuscript.

Reviewers' comments:

Reviewer #1 (Remarks to the Author):

The authors have added substantial amount of new data to the manuscript, which in my opinion is greatly improved. In particular, they have significantly strengthened the mechanistic link between Dll4 and EGFL7 and as such my concerns have now been addressed in full.

I would like to see a minor change to the manuscript before acceptance though, which concerns the new data on the EGFL7; nestinCreERT2 presented in the revised manuscript. The quantifications of TAPs and NBs strengthen the authors findings on the role of EGFL7 in NPC differentiation towards the neuronal lineage. However, in the absence of data on the qNSC/aNSC compartment, the finding also somewhat undermines the specific role of vascular EGFL7 on neurogenesis presented in the first part of the manuscript. In line 128 the authors state that the vasculature is likely to be the major source of EGFL7 in the niche and the manuscript hinges on the importance of vascular EGFL7 in controlling NSC quiescence. As such, it would be critical to show that in the nestinCreER line the effect on quiescent NSC is marginal or none. This would confirm that vascular EGFL7 maintains NSC quiescence, whereas neural EGFL7 regulates later stages of the lineage and further strengthen the paper overall. This should not be very labor-intensive as the authors are likely to already have SVZ tissue for this from their analysis of TAPs.

Reviewer #2 (Remarks to the Author):

The authors have addressed all of my initial concerns either by clarifying the text or adding new data. I feel that the revised manuscript has improved considerably and the data and interpretation now reflect the claims.

Reviewer #3 (Remarks to the Author):

The authors have added experiments to address some of the concerns, and the manuscript is improved. However, overall, it remains a very complicated manuscript that is difficult to follow, as there are so many different types of experiments that are performed somewhat superficially, with read-outs interpreted as demonstrating effects on different SVZ cell types, although the assays do not allow such specific interpretation.

The paper is framed as EGFL7 playing an important role in the vascular/stem cell niche. In fact the authors posit in the introduction that EGFL7 is a vascular derived factor. The authors now include cell type specific inducible deletion experiments to circumvent the issue of constitutive deletion through all of development. Unfortunately they do not directly compare deletion of EGFL7 in vascular cells and in neural progenitor cells (nestin+) cells.

The higher power images that are now included are easier to see, but are not especially convincing in showing co-labeling of EGFL7 and different cell SVZ types. Of note, qNSCs express very high levels of EGFL7 by qPCR, higher than endothelial cells. However, these cells are ignored throughout the in vivo analysis until Figure 5. Although the authors state that there are more endothelial cells in the SVZ than qNSCs (line 128) it is not clear what this statement is based on, and the high levels in qNSCs are likely to be relevant. In addition, Nestin+GFAP+ correspond to activated neural stem cells. They are referred to throughout the text as neural stem cells. GFAP+ nestin- cells contacting the ventricle are quiescent neural stem cells. Indeed when these are quantified in Figure 5 in the EGFL7^{-/-} mice, a

decrease in this population is seen.

A remaining issue is that different aspects of the NS assay are used to draw conclusions about in vivo populations. An important concern that has not been addressed in the revision is the behaviour of different FACS purified populations (qNSCs, aNSCs, TAPs, and early NBs) after the different manipulations. For example, Figure 2 uses secondary NS (size and number) as a readout. In Figure 3, they switch to primary spheres. It is difficult to compare the findings of Figure 2 (with secondary spheres) with Figure 3 (primary spheres). In Figure 5, they state that all NS initiating cells are activated NSCs. However, multiple populations give rise to primary NS, and all of which are dividing.

Time-lapse imaging: The authors state that cells that have appearance of quiescent astroglial cells re-enter cell cycle only in the mutant. It is unclear how these cells are being characterized. From other studies it is known that NSCs undergo several rounds of division. These cells cannot be classified as quiescent astroglial cells, without any markers. Without any controls or quantification of lineage trees, this analysis cannot be interpreted.

It is still somewhat perplexing as to why there are more activated stem cells, but fewer neurons migrating to the olfactory bulb in the EGFL7^{-/-} mice. What is happening to the progeny of the activated stem cells? Are more oligodendrocytes being generated by SVZ stem cells (ie more BrdU+ cells in the corpus callosum)? One important further consideration is whether the main olfactory epithelium is affected in EGFL7^{-/-} mice, which could also affect olfactory behavior.

Other comments:

Suppl Figure 2C - How does this figure show that NS formed after FACS are the same as at clonal density? This graph shows secondary NS formation.

In supplement 2C-F. what does the y axis in all of the graphs mean "Number of secondary spheres formed (%)". Is this number or %?

In Suppl Figure 1 M', N', asterisks refer to non-endothelial cells and arrowheads to endothelial cells. The figure legend should be corrected.

In the abstract, the authors still discuss metabolism, although do not present data on this.

Please find below our responses to the editor's and the reviewer's comments.

Reviewer #1 (Remarks to the Author):

The authors have added substantial amount of new data to the manuscript, which in my opinion is greatly improved. In particular, they have significantly strengthened the mechanistic link between Dll4 and EGFL7 and as such my concerns have now been addressed in full.

I would like to see a minor change to the manuscript before acceptance though, which concerns the new data on the EGFL7; nestinCreERT2 presented in the revised manuscript. The quantifications of TAPs and NBs strengthen the authors findings on the role of EGFL7 in NPC differentiation towards the neuronal lineage. However, in the absence of data on the qNSC/aNSC compartment, the finding also somewhat undermines the specific role of vascular EGFL7 on neurogenesis presented in the first part of the manuscript. In line 128 the authors state that the vasculature is likely to be the major source of EGFL7 in the niche and the manuscript hinges on the importance of vascular EGFL7 in controlling NSC quiescence. As such, it would be critical to show that in the nestinCreER line the effect on quiescent NSC is marginal or none. This would confirm that vascular EGFL7 maintains NSC quiescence, whereas neural EGFL7 regulates later stages of the lineage and further strengthen the paper overall. This should not be very labor-intensive as the authors are likely to already have SVZ tissue for this from their analysis of TAPs.

Response to the reviewer

In order to address Reviewer 1's concern we created a new mouse line allowing for the tamoxifen-inducible knock-out of EGFL7 in blood vessels ($EGFL7^{fl/fl}; Cdh5-CreERT2$). Subsequently, the amount of quiescent neural stem cells (qNSCs; $GFAP^+/Nestin^-/S100\beta^-$) in the subventricular zone of these animals has been determined and was compared to the amount of qNSCs counted in $EGFL7^{fl/fl}; Nestin-CreERT2$ mice upon the tamoxifen-induced

knock-out of EGFL7. Data revealed a significant reduction in qNSCs upon the blood vessel-specific knock-out of EGFL7 which was not observed upon EGFL7 knock-out in neural stem cells (new Fig. 5O-R).

Reviewer #2 (Remarks to the Author):

The authors have addressed all of my initial concerns either by clarifying the text or adding new data. I feel that the revised manuscript has improved considerably and the data and interpretation now reflect the claims.

Response to the reviewer

We are grateful for the contribution of Reviewer 2 to our manuscript.

Reviewer #3 (Remarks to the Author):

The authors have added experiments to address some of the concerns, and the manuscript is improved. However, overall, it remains a very complicated manuscript that is difficult to follow, as there are so many different types of experiments that are performed somewhat superficially, with read-outs interpreted as demonstrating effects on different SVZ cell types, although the assays do not allow such specific interpretation.

Response to the reviewer

Although we agree that the body of evidence presented in our manuscript grew fairly large, in part due to additional experiments that were requested during the revision process, we ensure Reviewer 3 that despite not presenting each experiment in extensive detail, none of the experiments were done in a superficial manner. All experiments were carefully designed and performed according to good scientific practice. The interpretation of all results has been done in a cautious and conservative manner. The manuscript has been screened once again for redundant or marginal data; however, no such data was found.

The paper is framed as EGFL7 playing an important role in the vascular/stem cell niche. In fact the authors posit in the introduction that EGFL7 is a vascular derived factor. The authors now include cell type specific inducible deletion experiments to circumvent the issue of constitutive deletion through all of development. Unfortunately they do not directly compare deletion of EGFL7 in vascular cells and in neural progenitor cells (nestin+) cells.

Response to the reviewer

In order to address Reviewer 3's concern, we compared the amount of qNSCs (GFAP⁺/Nestin⁻/S100β⁻) in mice specifically lacking EGFL7 in aNSCs/NPCs (*EGFL7^{fl/fl}; Nestin-*

CreERT2) or in blood vessels (*EGFL7^{fl/fl};Cdh5-CreERT2*). A reduction in qNSCs was only observed upon the blood vessel-specific knock-out of EGFL7 (new Figs. 5O-R), supporting our statement that vascular rather than neural EGFL7 affects NSC quiescence.

The higher power images that are now included are easier to see, but are not especially convincing in showing co-labeling of EGFL7 and different cell SVZ types.

Response to the reviewer

Unfortunately, the quality of the pictures cannot be further improved using the hardware available to us (confocal microscopes Leica SP5 and 8). However, we'd like to point out that the EGFL7 FISH-labeling applied in several of the pictures delivers a punctuated staining pattern. This looks very different from the more homogenous antibody staining patterns that have been used to identify different SVZ cell types. Putatively, the combination of the two methods, delivering quite different staining patterns, may give the false impression that the co-labelings shown in the manuscript are not convincing.

Of note, qNSCs express very high levels of EGFL7 by qPCR, higher than endothelial cells. However, these cells are ignored throughout the in vivo analysis until Figure 5. Although the authors state that there are more endothelial cells in the SVZ than qNSCs (line 128) it is not clear what this statement is based on, and the high levels in qNSCs are likely to be relevant.

Response to the reviewer

This statement was based on the total numbers of ECs and NSCs we retrieved from the SVZ upon FACS sorting. Typically, the ratio of ECs to NSCs was greater than 10 to 1. In order to address the reviewer's concern we added this piece of information to the manuscript. Certainly, we did not mean to brush off qNSCs in our manuscript by analyzing them in Figure 5. The order of Figures does not reflect our appreciation of certain pieces

of data but solely represents our attempt to structure the manuscript in a clear and convenient manner.

In addition, Nestin+GFAP+ correspond to activated neural stem cells. They are referred to throughout the text as neural stem cells. GFAP+ nestin- cells contacting the ventricle are quiescent neural stem cells. Indeed when these are quantified in Figure 5 in the EGL7-/-mice, a decrease in this population is seen.

Response to the reviewer

This oversimplification, which was solely applied to increase the readability of the text, has been corrected throughout the manuscript.

A remaining issue is that different aspects of the NS assay are used to draw conclusions about in vivo populations. An important concern that has not been addressed in the revision is the behaviour of different FACS purified populations (qNSCs, aNSCs, TAPs, and early NBs) after the different manipulations. For example, Figure 2 uses secondary NS (size and number) as a readout. In Figure 3, they switch to primary spheres. It is difficult to compare the findings of Figure 2 (with secondary spheres) with Figure 3 (primary spheres). In Figure 5, they state that all NS initiating cells are activated NSCs. However, multiple populations give rise to primary NS, and all of which are dividing.

Response to the reviewer

The reviewer's point is well taken; however, we did not directly compare *in vitro* and *in vivo* NSC populations. Actually, the manuscript is in part about the fact that this cannot easily be done and we elaborate on this topic intensively in the discussion. The neurosphere assay (Fig. 2) serves as a model to analyze the role of neural EGFL7 in neurosphere formation. In the absence of any cells except NSCs and their progeny all

EGFL7 is derived from these cells and only these cellular interfaces matter. So we can use the neurosphere assay *in vitro* to learn something about the impact of “neural” EGFL7 on neural stem cells and their progeny. However, *in vivo* additional cell types exist, e.g., the endothelial cells in blood vessels, which produce high levels of “vascular” EGFL7 and this EGFL7 pool needs to be considered as well. Neural and vascular EGFL7 must not necessarily have identical functions and, as we show in the manuscript, they do not. Unfortunately, we cannot isolate primary neurospheres (Fig. 3 and 5) and subsequently culture them to compare secondary sphere formation as done in Fig. 2. In this case we would eliminate the neurovascular interface and only retain the neural interactions, so it would be the same assay as in Fig. 2. Therefore, we use the primary sphere formation assay to assess how many cells in the SVZ were able to form a sphere *in vivo* at the time when the mice were sacrificed. However, we did not speculate on the identity of these neurosphere-initiating cells, i.e., whether or not these cells are activated NSCs, TAPs or neuroblasts. We solely concluded that upon the loss of EGFL7 *in vivo* more cells in the SVZ are able to form a neurosphere.

Nevertheless, we established the assay requested by the reviewer and sorted SVZ tissue derived from EGFL7^{-/-} and WT mice for qNSCs, aNSCs, TAPs and neuroblasts. In particular, the neurosphere formation capacity of aNSCs was increased upon the loss of EGFL7, while the TAPs were not significantly affected (new Fig. 2F). qNSCs and NBs formed no (or only a few) neurospheres.

Time-lapse imaging: The authors state that cells that have appearance of quiescent astroglial cells re-enter cell cycle only in the mutant. It is unclear how these cells are being characterized. From other studies it is known that NSCs undergo several rounds of division. These cells cannot be classified as quiescent astroglial cells, without any markers. Without any controls or quantification of lineage trees, this analysis cannot be interpreted.

Response to the reviewer

Conclusions drawn in the time-lapse imaging assay are based on the expertise of our collaboration partners (and study coauthors) Benedikt Berninger and Felipe Ortega, who are very experienced in it and have published the assay in high ranking journals (Costa and Ortega et al., *Development*, 2011; Ortega and Costa et al., *Nat Protocols*, 2011). A major benefit of this *in vitro* culture model is that NSCs differentiate in a quite standardized manner, which allows for the calculation of accurate percentages of neurogenic trees. Active NSCs are defined by some major hallmarks in this assay: GFAP expression, slow cell cycle, massive growth prior to division, and the capability of performing asymmetric cell divisions. Thereby, the slowly dividing aNSCs give rise to fast dividing astroglia that progress asymmetrically with one branch giving rise to TAPs and neurons, whereas the other gives rise again to quiescent GFAP⁺ cells. Usually, these quiescent GFAP⁺ cells only re-enter the cell cycle if challenged by strong mitotic factors such as EGF or FGF2. The analysis of more than 1,500 clones in various experimental settings confirmed that no relevant percentage of quiescent GFAP⁺ cells ever re-entered the cell cycle in the absence of mitogens (the experimental setting in the manuscript). This only occurred if the assay was performed using EGFL7^{-/-} NSCs. Cells first gave rise to two prototypical flat and large GFAP⁺ cells, which are considered quiescent in the manuscript but after 2-3 days re-entered the cell cycle and gave rise to TAPs and neurons. This finding is depicted in Fig. 2G and can be deduced from the tree library in subfigure I. In order to illustrate this finding we added pictures, movies and an illustration. Furthermore, the effect of EGFL7^{-/-} on NSC activation is not only seen in this assay but is a recurring motif throughout the manuscript.

It is still somewhat perplexing as to why there are more activated stem cells, but fewer neurons migrating to the olfactory bulb in the EGFL7^{-/-} mice. What is happening to the progeny of the activated stem cells? Are more oligodendrocytes being generated by SVZ stem cells (ie more BrdU+ cells in the corpus callosum)?

[UNPUBLISHED DATA REDACTED BY EDITORIAL TEAM AS PER AUTHOR REQUEST]

One important further consideration is whether the main olfactory epithelium is affected in EGFL7^{-/-} mice, which could also affect olfactory behavior.

Response to the reviewer

In order to address the reviewer's concern, the surface of the olfactory epithelium has been quantified in both EGFL7^{-/-} and wild-type mice. However, no difference was observed between the groups (new Suppl. Fig. 6O-Q).

Other

comments:

Suppl Figure 2C- How does this figure show that NS formed after FACS are the same as at clonal density? This graph shows secondary NS formation.

Response to the reviewer

This graph illustrates that NSC self-renewal data obtained under clonal conditions (Suppl. Fig. 2C) or at clonal density (Suppl. Fig. 2D+F) were comparable and that both techniques can be applied to collect this data.

In supplement 2C-F. what does the y axis in all of the graphs mean “Number of secondary spheres formed (%)”. Is this number or %?

Response to the reviewer

The labeling of the y axes has been adjusted in the new version of the manuscript.

In Suppl Figure 1 M', N', asterisks refer to non-endothelial cells and arrowheads to endothelial cells. The figure legend should be corrected.

Response to the reviewer

The mislabeling has now been corrected.

In the abstract, the authors still discuss metabolism, although do not present data on this.

Response to the reviewer

The term metabolism has been erased from the abstract of the re-revised version of the manuscript.

REVIEWERS' COMMENTS:

Reviewer #1 (Remarks to the Author):

All of my concerns have been addressed in full by the additional data and the conclusions are well supported by the results.

Reviewer #3 (Remarks to the Author):

The authors have addressed my previous concerns with new experiments and text changes. The revised manuscript is greatly improved and data and text now reflect their claims.